# MC-LSTM: Mass-Conserving LSTM

## Abstract

The success of Convolutional Neural Networks (CNNs) in computer vision is mainly driven by their strong inductive bias, which is strong enough to allow CNNs to solve vision-related tasks with random weights, meaning without learning. Similarly, Long Short-Term Memory (LSTM) has a strong inductive bias towards storing information over time. However, many real-world systems are governed by conservation laws, which lead to the redistribution of particular quantities — e.g. in physical and economical systems. Our novel Mass-Conserving LSTM (MC-LSTM) adheres to these conservation laws by extending the inductive bias of LSTM to model the redistribution of those stored quantities. MC-LSTMs set a new state-of-the-art for neural arithmetic units at learning arithmetic operations, such as addition tasks, which have a strong conservation law, as the sum is constant over time. Further, MC-LSTM is applied to traffic forecasting, modeling a pendulum, and a large benchmark dataset in hydrology, where it sets a new state-of-the-art for predicting peak flows. In the hydrology example, we show that MC-LSTM states correlate with real world processes and are therefore interpretable.

## 1 Introduction

**Inductive biases enabled the success of CNNs and LSTMs.** One of the greatest success stories of deep learning is Convolutional Neural Networks (CNNs) (Fukushima, 1980; LeCun & Bengio, 1998; Schmidhuber, 2015; LeCun et al., 2015) whose proficiency can be attributed to their strong inductive bias towards visual tasks (Cohen & Shashua, 2017; Gaier & Ha, 2019). The effect of this inductive bias has been demonstrated by CNNs that solve vision-related tasks with random weights, meaning without learning (He et al., 2016; Gaier & Ha, 2019; Ulyanov et al., 2020). Another success story is Long Short-Term Memory (LSTM) (Hochreiter, 1991; Hochreiter & Schmidhuber, 1997), which has a strong inductive bias toward storing information through its memory cells. This inductive bias allows LSTM to excel at speech, text, and language tasks (Sutskever et al., 2014; Bohnet et al., 2018; Kochkina et al., 2017; Liu & Guo, 2019), as well as timeseries prediction. Even with random weights and only a learned linear output layer LSTM is better at predicting timeseries than reservoir methods (Schmidhuber et al., 2007). In a seminal paper on biases in machine learning, Mitchell (1980) stated that *"biases and initial knowledge are at the heart of the ability to generalize beyond observed data"*. Therefore, choosing an appropriate architecture and inductive bias for deep neural networks is key to generalization.

**Mechanisms beyond storing are required for real-world applications.** While LSTM can store information over time, real-world applications require mechanisms that go beyond storing. Many real-world systems are governed by conservation laws related to mass, energy, momentum, charge, or particle counts, which are often expressed through continuity equations. In physical systems, different types of energies, mass or particles have to be conserved (Evans & Hanney, 2005; Rabitz et al., 1999; van der Schaft et al., 1996), in hydrology it is the amount of water (Freeze & Harlan, 1969; Beven, 2011), in traffic and transportation the number of vehicles (Vanajakshi & Rilett, 2004; Xiao & Duan, 2020; Zhao et al., 2017), and in logistics the amount of goods, money or products. A real-world task could be to predict outgoing goods from a warehouse based on a general state of the warehouse, i.e., how many goods are in storage, and incoming supplies. If the predictions are not precise, then they do not lead to an optimal control of the production process. For modeling such systems, certain inputs must be conserved but also redistributed across storage locations within the system. We will

---

All code to reproduce the results will be made available on GitHub.

refer to conserved inputs as *mass*, but note that this can be any type of conserved quantity. We argue that for modeling such systems, specialized mechanisms should be used to represent locations & whereabouts, objects, or storage & placing locations and thus enable conservation.

**Conservation laws should pervade machine learning models in the physical world.** Since a large part of machine learning models are developed to be deployed in the real world, in which conservation laws are omnipresent rather than the exception, these models should adhere to them automatically and benefit from them. However, standard deep learning approaches struggle at conserving quantities across layers or timesteps (Beucler et al., 2019b; Greydanus et al., 2019; Song & Hopke, 1996; Yitian & Gu, 2003), and often solve a task by exploiting spurious correlations (Szegedy et al., 2014; Lapuschkin et al., 2019). Thus, an inductive bias of deep learning approaches via mass conservation over time in an open system, where mass can be added and removed, could lead to a higher generalization performance than standard deep learning for the above-mentioned tasks.

**A mass-conserving LSTM.** In this work, we introduce Mass-Conserving LSTM (MC-LSTM), a variant of LSTM that enforces mass conservation by design. MC-LSTM is a recurrent neural network with an architecture inspired by the gating mechanism in LSTMs. MC-LSTM has a strong inductive bias to guarantee the conservation of mass. This conservation is implemented by means of left-stochastic matrices, which ensure the sum of the memory cells in the network represents the current mass in the system. These left-stochastic matrices also enforce the mass to be conserved through time. The MC-LSTM gates operate as control units on mass flux. Inputs are divided into a subset of *mass inputs*, which are propagated through time and are conserved, and a subset of *auxiliary inputs*, which serve as inputs to the gates for controlling mass fluxes. We demonstrate that MC-LSTMs excel at tasks where conservation of mass is required and that it is highly apt at solving real-world problems in the physical domain.

**Contributions.** We propose a novel neural network architecture based on LSTM that conserves quantities, such as mass, energy, or count, of a specified set of inputs. We show properties of this novel architecture, called MC-LSTM, and demonstrate that these properties render it a powerful neural arithmetic unit. Further, we show its applicability in real-world areas of traffic forecasting and modeling the pendulum. In hydrology, large-scale benchmark experiments reveal that MC-LSTM has powerful predictive quality and can supply interpretable representations.

## 2 MASS-CONSERVING LSTM

The original LSTM introduced memory cells to Recurrent Neural Networks (RNNs), which alleviate the vanishing gradient problem (Hochreiter, 1991). This is achieved by means of a fixed recurrent self-connection of the memory cells. If we denote the values in the memory cells at time $t$ by $\boldsymbol{c}^t$, this recurrence can be formulated as

$$\boldsymbol{c}^t = \boldsymbol{c}^{t-1} + f(\boldsymbol{x}^t, \boldsymbol{h}^{t-1}), \tag{1}$$

where $\boldsymbol{x}$ and $\boldsymbol{h}$ are, respectively, the forward inputs and recurrent inputs, and $f$ is some function that computes the increment for the memory cells. Here, we used the original formulation of LSTM without forget gate (Hochreiter & Schmidhuber, 1997), but in all experiments we also consider LSTM with forget gate (Gers et al., 2000).

MC-LSTMs modify this recurrence to guarantee the conservation of the mass input. The key idea is to use the memory cells from LSTMs as mass accumulators, or mass storage. The conservation law is implemented by three architectural changes. First, the increment, computed by $f$ in Eq. (1), has to distribute mass from inputs into accumulators. Second, the mass that leaves MC-LSTM must also disappear from the accumulators. Third, mass has to be redistributed between mass accumulators. These changes mean that all gates explicitly represent mass fluxes.

Since, in general, not all inputs must be conserved, we distinguish between *mass* inputs, $\boldsymbol{x}$, and *auxiliary* inputs, $\boldsymbol{a}$. The former represents the quantity to be conserved and will fill the mass accumulators in MC-LSTM. The auxiliary inputs are used to control the gates. To keep the notation uncluttered, and without loss of generality, we use a single mass input at each timestep, $x^t$, to introduce the architecture.

The forward pass of MC-LSTM at timestep $t$ can be specified as follows:

$$m_{\text{tot}}^t = \boldsymbol{R}^t \cdot \boldsymbol{c}^{t-1} + \boldsymbol{i}^t \cdot x^t \qquad (2)$$

$$\boldsymbol{c}^t = (\boldsymbol{1} - \boldsymbol{o}^t) \odot m_{\text{tot}}^t \qquad (3)$$

$$\boldsymbol{h}^t = \boldsymbol{o}^t \odot m_{\text{tot}}^t. \qquad (4)$$

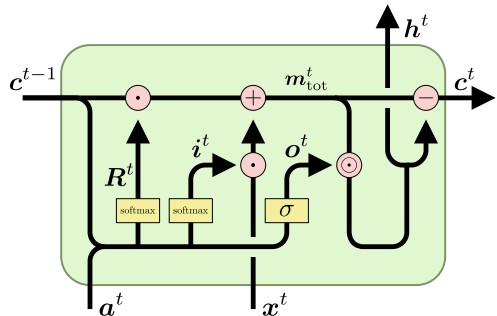

Figure 1: Schematic representation of the main operations in the MC-LSTM architecture (adapted from: Olah, 2015).

where $\boldsymbol{i}^t$ and $\boldsymbol{o}^t$ are the input- and output gates, respectively, and $\boldsymbol{R}$ is a positive left-stochastic matrix, i.e., $\boldsymbol{1}^T \cdot \boldsymbol{R} = \boldsymbol{1}$, for redistributing mass in the accumulators. The *total mass* $m_{\text{tot}}$ is the *redistributed mass*, $\boldsymbol{R}^t \cdot \boldsymbol{c}^{t-1}$, plus the *mass influx*, or new mass, $\boldsymbol{i}^t \cdot x^t$. The current mass in the system is stored in $\boldsymbol{c}^t$.

Note the differences between Eq. (1) and Eq. (3). First, the increment of the memory cells no longer depends on $\boldsymbol{h}^t$. Instead, mass inputs are distributed by means of the normalized $\boldsymbol{i}$ (see Eq. 5). Furthermore, $\boldsymbol{R}^t$ replaces the implicit identity matrix of LSTM to redistribute mass among memory cells. Finally, Eq. (3) introduces $\boldsymbol{1} - \boldsymbol{o}^t$ as a forget gate on the total mass, $m_{\text{tot}}$. Together with Eq. (4), this assures that no outgoing mass is stored in the accumulators. This formulation has some similarity to Gated Recurrent Units (GRU) (Cho et al., 2014), however the gates are not used for mixing the old and new cell state, but for splitting off the output.

**Basic gating and redistribution.** The MC-LSTM gates at timestep $t$ are computed as follows:

$$\boldsymbol{i}^t = \text{softmax}\left(\boldsymbol{W}_{\text{i}} \cdot \boldsymbol{a}^t + \boldsymbol{U}_{\text{i}} \cdot \frac{\boldsymbol{c}^{t-1}}{\|\boldsymbol{c}^{t-1}\|_1} + \boldsymbol{b}_{\text{i}}\right) \qquad (5)$$

$$\boldsymbol{o}^t = \sigma\left(\boldsymbol{W}_{\text{o}} \cdot \boldsymbol{a}^t + \boldsymbol{U}_{\text{o}} \cdot \frac{\boldsymbol{c}^{t-1}}{\|\boldsymbol{c}^{t-1}\|_1} + \boldsymbol{b}_{\text{o}}\right) \qquad (6)$$

$$\boldsymbol{R}^t = \text{softmax}(\boldsymbol{B}_{\text{r}}), \qquad (7)$$

where the $\text{softmax}$ operator is applied column-wise, $\sigma$ is the logistic sigmoid function, and $\boldsymbol{W}_{\text{i}}$, $\boldsymbol{b}_{\text{i}}$, $\boldsymbol{W}_{\text{o}}$, $\boldsymbol{b}_{\text{o}}$, and $\boldsymbol{B}_{\text{r}}$ are learnable model parameters. Note that for the input gate and redistribution matrix, the requirement is that they are column normalized. This can also be achieved by other means than using the softmax function. For example, an alternative way to ensure a column-normalized matrix $\boldsymbol{R}^t$ is to use a normalized logistic, $\tilde{\sigma}(r_{kj}) = \frac{\sigma(r_{kj})}{\sum_n \sigma(r_{kn})}$. Also note that MC-LSTMs compute the gates from the memory cells, directly. This is in contrast with the original LSTM, which uses the activations from the previous time step. The accumulated values from the memory cells, $\boldsymbol{c}^t$, are normalized to counter saturation of the sigmoids and to supply probability vectors that represent the current distribution of the mass across cell states We use this variation e.g. in our experiments with *neural arithmetics* (see Sec. 5.1).

**Time-dependent redistribution.** It can also be useful to predict a redistribution matrix for each sample and timestep, similar to how the gates are computed:

$$\boldsymbol{R}^t = \text{softmax}\left(\mathbf{W}_{\text{r}} \cdot \boldsymbol{a}^t + \mathbf{U}_{\text{r}} \cdot \frac{\boldsymbol{c}^{t-1}}{\|\boldsymbol{c}^{t-1}\|_1} + \boldsymbol{B}_{\text{r}}\right), \qquad (8)$$

where the parameters $\mathbf{W}_{\text{r}}$ and $\mathbf{U}_{\text{r}}$ are weight tensors and their multiplications result in $K \times K$ matrices. Again, the $\text{softmax}$ function is applied column-wise. This version collapses to a time-independent redistribution matrix if $\mathbf{W}_{\text{r}}$ and $\mathbf{U}_{\text{r}}$ are equal to $\boldsymbol{0}$. Thus, there exists the option to initialize $\mathbf{W}_{\text{r}}$ and $\mathbf{U}_{\text{r}}$ with weights that are small in absolute value compared to the weights of $\boldsymbol{B}_{\text{r}}$, to favour learning time-independent redistribution matrices. We use this variant in the hydrology experiments (see Sec. 5.4).

**Redistribution via a hypernetwork.** Even more general, a hypernetwork (Schmidhuber, 1992; Ha et al., 2017) that we denote with $g$ can be used to procure $\boldsymbol{R}$. The hypernetwork has to produce

a column-normalized, square matrix $\boldsymbol{R}^t = g(\boldsymbol{a}^0, \ldots, \boldsymbol{a}^t, \boldsymbol{c}^0, \ldots, \boldsymbol{c}^{t-1})$. Notably, a hypernetwork can be used to design an *autoregressive* version of MC-LSTMs, if the network additionally predicts auxiliary inputs for the next time step. We use this variant in the pendulum experiments (see Sec. 5.3).

## 3 PROPERTIES

**Conservation.**  MC-LSTM guarantees that mass is conserved over time. This is a direct consequence of connecting memory cells with stochastic matrices. The mass conservation ensures that no mass can be removed or added implicitly, which makes it easier to learn functions that generalize well. The exact meaning of this mass conservation is formalized in Theorem 1.

**Theorem 1** (Conservation property). *Let* $m_c^\tau = \sum_{k=1}^K c_k^\tau$ *be the mass contained in the system and* $m_h^\tau = \sum_{k=1}^K h_k^\tau$ *be the mass efflux, or, respectively, the* accumulated mass *in the MC-LSTM storage and the outputs at time* $\tau$. *At any timestep* $\tau$, *we have:*

$$m_c^\tau = m_c^0 + \sum_{t=1}^\tau x^t - \sum_{t=1}^\tau m_h^t. \tag{9}$$

*That is, the change of mass in the memory cells is the difference between the input and output mass, accumulated over time.*

The proof is by induction over $\tau$ (see Appendix C). Note that it is still possible for input mass to be stored indefinitely in a memory cell so that it does not appear at the output. This can be a useful feature if not all of the input mass is needed at the output. In this case, the network can learn that one cell should operate as a collector for excess mass in the system.

**Boundedness of cell states.**  In each timestep $\tau$, the memory cells, $c_k^\tau$, are bounded by the sum of mass inputs $\sum_{t=1}^\tau x^t + m_c^0$, that is $|c_k^\tau| \leq \sum_{t=1}^\tau x^t + m_c^0$. Furthermore, if the series of mass inputs converges, $\lim_{\tau \to \infty} \sum_{t=1}^\tau x^\tau = m_x^\infty$, then also the sum of cell states converges (see Appendix, Corollary 1).

**Initialization and gradient flow.**  MC-LSTM with $\boldsymbol{R}^t = \boldsymbol{I}$ has a similar gradient flow to LSTM with forget gate (Gers et al., 2000). Thus, the main difference in the gradient flow is determined by the redistribution matrix $\boldsymbol{R}$. The forward pass of MC-LSTM without gates $\boldsymbol{c}^t = \boldsymbol{R}^t \boldsymbol{c}^{t-1}$ leads to the following backward expression $\frac{\partial \boldsymbol{c}^t}{\partial \boldsymbol{c}^{t-1}} = \boldsymbol{R}^t$. Hence, MC-LSTM should be initialized with a redistribution matrix close to the identity matrix to ensure a stable gradient flow as in LSTMs. For random redistribution matrices, the *circular law theorem for random Markov matrices* (Bordenave et al., 2012) can be used to analyze the gradient flow in more detail, see Appendix, Section D.

**Computational complexity.**  Whereas the gates in a traditional LSTM are vectors, the input gate and redistribution matrix of an MC-LSTM are matrices in the most general case. This means that MC-LSTM is, in general, computationally more demanding than LSTM. Concretely, the forward pass for a single timestep in MC-LSTM requires $\mathcal{O}(K^3 + K^2(M + L) + KML)$ Multiply-Accumulate operations (MACs), whereas LSTM takes $\mathcal{O}(K^2 + K(M + L))$ MACs per timestep. Here, $M$, $L$ and $K$ are the number of mass inputs, auxiliary inputs and outputs, respectively. When using a time-independent redistribution matrix cf. Eq. (7), the complexity reduces to $\mathcal{O}(K^2 M + KML)$ MACs.

**Potential interpretability through inductive bias and accessible mass in cell states.**  The representations within the model can be interpreted directly as accumulated mass. If one mass or energy quantity is known, the MC-LSTM architecture would allow to force a particular cell state to represent this quantity, which could facilitate learning and interpretability. An illustrative example is the case of rainfall runoff modelling, where observations, say of the soil moisture or groundwater-state, could be used to guide the learning of an explicit memory cell of MC-LSTM.

## 4 SPECIAL CASES AND RELATED WORK

**Relation to Markov chains.** In a special case MC-LSTM collapses to a *finite Markov chain*, when $c^0$ is a probability vector, the mass input is zero $x^t = 0$ for all $t$, there is no input and output gate, and the redistribution matrix is constant over time $R^t = R$. For finite Markov chains, the dynamics are known to converge, if $R$ is irreducible (see e.g. Hairer (2018, Theorem 3.13.)). Awiszus & Rosenhahn (2018) aim to model a Markov Chain by having a feed-forward network predict the state distribution given the current state distribution. In order to insert randomness to the network, a random seed is appended to the input, which allows to simulate Markov processes. Although MC-LSTMs are closely related to Markov chains, they do not explicitly learn the transition matrix, as is the case for Markov chain neural networks. MC-LSTMs would have to learn the transition matrix implicitly.

**Relation to normalizing flows and volume-conserving neural networks.** In contrast to *normalizing flows* (Rezende & Mohamed, 2015; Papamakarios et al., 2019), which transform inputs in each layer and trace their density through layers or timesteps, MC-LSTMs transform distributions and do not aim to trace individual inputs through timesteps. Normalizing flows thereby conserve information about the input in the first layer and can use the inverted mapping to trace an input back to the initial space. MC-LSTMs are concerned with modeling the changes of the initial distribution over time and can guarantee that a multinomial distribution is mapped to a multinomial distribution. For MC-LSTMs without gates, the sequence of cell states $c^0, \ldots, c^T$ constitutes a *normalizing flow* if an initial distribution $p_0(c^0)$ is available. In more detail, MC-LSTM can be considered a *linear flow* with the mapping $c^{t+1} = R^t c^t$ and $p(c^{t+1}) = p(c^t) |\det R^t|^{-1}$ in this case. The gate providing the redistribution matrix (see Eq. 8) is the *conditioner* in a normalizing flow model. From the perspective of normalizing flows, MC-LSTM can be considered as a flow trained in a supervised fashion. Deco & Brauer (1995) proposed volume-conserving neural networks, which conserve the volume spanned by input vectors and thus the information of the starting point of an input is kept. In other words, they are constructed so that the Jacobians of the mapping from one layer to the next have a determinant of 1. In contrast, the MC-LSTMs determinant of the Jacobians (of the mapping) is smaller than 1 (except for degenerate cases), which means that volume of the inputs is not conserved.

**Relation to Layer-wise Relevance Propagation.** Layer-wise Relevance Propagation (LRP) (Bach et al., 2015) is similar to our approach with respect to the idea that the sum of a quantity, the relevance $Q^l$ is conserved over layers $l$. LRP aims to maintain the sum of the relevance values $\sum_{k=1}^{I} Q_i^{l-1} = \sum_{k=1}^{I} Q_i^{l-1}$ backward through a classifier in order to a obtain relevance values for each input feature.

**Relation to other networks that conserve particular properties.** While a standard feed-forward neural network does not give guarantees aside from the conservation of the proximity of datapoints through the continuity property. The *conservation of the first moments of the data distribution* in the form of normalization techniques (Ioffe & Szegedy, 2015) has had tremendous success. Here, batch normalization (Ioffe & Szegedy, 2015) could exactly conserve mean and variance across layers, whereas self-normalization (Klambauer et al., 2017) conserves those approximately. The *conservation of the spectral norm* of each layer in the forward pass has enabled the stable training of generative adversarial networks (Miyato et al., 2018). The *conservation of the spectral norm of the errors* through the backward pass of an RNN has enabled the avoidance of the vanishing gradient problem (Hochreiter, 1991; Hochreiter & Schmidhuber, 1997). In this work, we explore an architecture that exactly *conserves the mass of a subset of the input*, where mass is defined as a physical quantity such as mass or energy.

**Relation to neural networks for physical systems.** Neural networks have been shown to discover physical concepts such as the conservation of energies (Iten et al., 2020), and neural networks could allow to learn natural laws from observations (Schmidt & Lipson, 2009; Cranmer et al., 2020b). MC-LSTM can be seen as a neural network architecture with physical constraints (Karpatne et al., 2017; Beucler et al., 2019c). It is however also possible to impose conservation laws by using other means, e.g. initialization, constrained optimization or soft constraints (as, for example, proposed by Karpatne et al., 2017; Beucler et al., 2019c;a; Jia et al., 2019). Hamiltonian neural networks (Greydanus et al., 2019) and Symplectic Recurrent Neural Networks make energy conserving predictions by using the Hamiltonian (Chen et al., 2019), a function that maps the inputs to the quantity that

Table 1: Performance of different models on the LSTM addition task in terms of the MSE. MC-LSTM significantly (all $p$-values below .05) outperforms its competitors, LSTM (with high initial forget gate bias), NALU and NAU. Error bars represent 95%-confidence intervals across 100 runs.

| | reference[a] | seq length[b] | input range[c] | count[d] | combo[e] | NaN[f] |
|---|---|---|---|---|---|---|
| MC-LSTM | **0.004** $\pm$ 0.003 | **0.009** $\pm$ 0.004 | **0.8** $\pm$ 0.5 | **0.6** $\pm$ 0.4 | **4.0** $\pm$ 2.5 | 0 |
| LSTM | 0.008 $\pm$ 0.003 | 0.727 $\pm$ 0.169 | 21.4 $\pm$ 0.6 | 9.5 $\pm$ 0.6 | 54.6 $\pm$ 1.0 | 0 |
| NALU | 0.060 $\pm$ 0.008 | 0.059 $\pm$ 0.009 | 25.3 $\pm$ 0.2 | 7.4 $\pm$ 0.1 | 63.7 $\pm$ 0.6 | 93 |
| NAU | 0.248 $\pm$ 0.019 | 0.252 $\pm$ 0.020 | 28.3 $\pm$ 0.5 | 9.1 $\pm$ 0.2 | 68.5 $\pm$ 0.8 | 24 |

[a] training regime: summing 2 out of 100 numbers between 0 and 0.5.
[b] longer sequence lengths: summing 2 out of 1 000 numbers between 0 and 0.5.
[c] more *mass* in the input: summing 2 out of 100 numbers between 0 and 5.0.
[d] higher number of summands: summing 20 out of 100 numbers between 0 and 0.5.
[e] combination of previous scenarios: summing 10 out of 500 numbers between 0 and 2.5.
[f] Number of runs that did not converge.

needs to be conserved. By using the symplectic gradients, it is possible to move around in the input space, without changing the output of the Hamiltonian. Lagrangian Neural Networks (Cranmer et al., 2020a), extend the Hamiltonian concept by making it possible to use arbitrary coordinates as inputs.

All of these approaches, while very promising, assume closed physical systems and are thus to restrictive for the application we have in mind. Raissi et al. (2019) propose to enforce physical constraints on simple feed-forward networks by computing the partial derivatives with respect to the inputs and computing the partial differential equations explicitly with the resulting terms. This approach, while promising, does require an exact knowledge of the governing equations. By contrast, our approach is able to learn its own representation of the underlying process, while obeying the pre-specified conservation properties.

## 5 EXPERIMENTS

In the following, we discuss the experiments we conducted to demonstrate the broad applicability and high predictive performance of MC-LSTM in settings where mass conservation is required. For more details on the datasets and hyperparameter selection for each experiment, we refer to Appendix B.

### 5.1 ARITHMETIC TASKS

**Addition problem.** We first considered a problem for which exact mass conservation is required. One example for such a problem has been described in the original LSTM paper (Hochreiter & Schmidhuber, 1997), showing that LSTM is capable of summing two arbitrarily marked elements in a sequence of random numbers. We show that MC-LSTM is able to solve this task, but also generalizes better to longer sequences, input values in a different range and more summands. Table 1 summarizes the results of this method comparison and shows that MC-LSTM significantly outperformed the other models on all tests ($p$-value $\leq 0.03$, Wilcoxon test). In Appendix B.1.5, we provide a qualitative analysis of the learned model behavior for this task.

**Recurrent arithmetic.** Following Madsen & Johansen (2020), the inputs for this task are sequences of vectors, uniformly drawn from $[1, 2]^{10}$. For each vector in the sequence, the sum over two random subsets is calculated. Those values are then summed over time, leading to two values. The target output is obtained by applying the arithmetic operation to these two values. The auxiliary input for MC-LSTM is a sequence of ones, where the last element is $-1$ to signal the end of the sequence.

We evaluated MC-LSTM against NAUs and Neural Accumulators (NACs) directly in the framework of Madsen & Johansen (2020). NACs and NAUs use the architecture as presented in (Madsen & Johansen, 2020). That is, a single hidden layer with two neurons, where the first layer is recurrent. The MC-LSTM model has two layers, of which the second one is a fully connected linear layer. For subtraction an extra cell was necessary to properly discard redundant input mass.

Table 2: Recurrent arithmetic task results. MC-LSTMs for addition and subtraction/multiplication have two and three neurons, respectively. Error bars represent 95%-confidence intervals.

| | addition | | subtraction | | multiplication | |
|---|---|---|---|---|---|---|
| | success rate[a] | updates[b] | success rate[a] | updates[b] | success rate[a] | updates[b] |
| MC-LSTM | $\mathbf{96}\%\ ^{+2\%}_{-6\%}$ | $4.6\cdot10^5$ | $81\%\ ^{+6\%}_{-9\%}$ | $1.2\cdot10^5$ | $\mathbf{67}\%\ ^{+8\%}_{-10\%}$ | $1.8\cdot10^5$ |
| NAU / NMU | $88\%\ ^{+5\%}_{-8\%}$ | $8.1\cdot10^4$ | $60\%\ ^{+9\%}_{-10\%}$ | $6.1\cdot10^4$ | $34\%\ ^{+10\%}_{-9\%}$ | $8.5\cdot10^4$ |
| NAC | $56\%\ ^{+9\%}_{-10\%}$ | $3.2\cdot10^5$ | $\mathbf{86}\%\ ^{+5\%}_{-8\%}$ | $4.5\cdot10^4$ | $0\%\ ^{+4\%}_{-0\%}$ | – |
| NALU | $10\%\ ^{+7\%}_{-4\%}$ | $1.0\cdot10^6$ | $0\%\ ^{+4\%}_{-0\%}$ | – | $1\%\ ^{+4\%}_{-1\%}$ | $4.3\cdot10^5$ |

[a] Percentage of runs that generalized to longer sequences.
[b] Median number of updates necessary to solve the task.

For testing, we took the model with the lowest validation error, c.f. early stopping. The performance was measured by the percentage of runs that successfully generalized to longer sequences. Generalization is considered successful if the error is lower than the numerical imprecision of the exact operation (Madsen & Johansen, 2020). The summary in Tab. 2 shows that MC-LSTM was able to significantly outperform the competing models ($p$-value 0.03 for addition and $3e-6$ for multiplication, proportion test). In Appendix B.1.5, we provide a qualitative analysis of the learned model behavior for this task.

**Static arithmetic.** To enable a direct comparison with the results reported in Madsen & Johansen (2020), we also compared MC-LSTM on the static arithmetic task, see Appendix B.1.3.

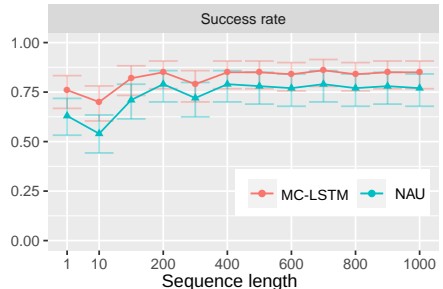

Figure 2: MNIST arithmetic task results for MC-LSTM and NAU. The task is to correctly predict the sum of a sequence of presented MNIST digits. The success rates are depicted on the y-axis in dependency of the length of the sequence (x-axis) of MNIST digits. Error bars represent 95%-confidence intervals.

**MNIST arithmetic.** We tested that feature extractors can be learned from MNIST images (LeCun et al., 1998) to perform arithmetic on the images (Madsen & Johansen, 2020). The input is a sequence of MNIST images and the target output is the corresponding sum of the labels. Auxiliary inputs are all 1, except the last entry, which is $-1$, to indicate the end of the sequence. The models are the same as in the recurrent arithmetic task with CNN to convert the images to (mass) inputs for these networks. The network is learned end-to-end. $L_2$-regularization is added to the output of CNN to prevent its outputs from growing arbitrarily large. The results for this experiment are depicted in Fig. 2. MC-LSTM significantly outperforms the state-of-the-art, NAU ($p$-value 0.002, Binomial test).

## 5.2 INBOUND-OUTBOUND TRAFFIC FORECASTING

We examined the usage of MC-LSTMs for traffic forecasting in situations in which inbound and outbound traffic counts of a city are available (see Fig. 3). For this type of data, a *conservation-of-vehicles* principle (Nam & Drew, 1996) must hold, since vehicles can only leave the city if they have entered it before or had been there in the first place. Based on data for the traffic4cast 2020 challenge (Kreil et al., 2020), we constructed a dataset to model inbound and outbound traffic in three different cities: Berlin, Istanbul and Moscow. We compared MC-LSTM against LSTM, which is the state-of-the-art method for several types of traffic forecasting situations (Zhao et al., 2017; Tedjopurnomo et al., 2020), and found that MC-LSTM significantly outperforms LSTM in this traffic forecasting setting (all $p$-values $\leq 0.01$, Wilcoxon test). For details, see Appendix B.2.

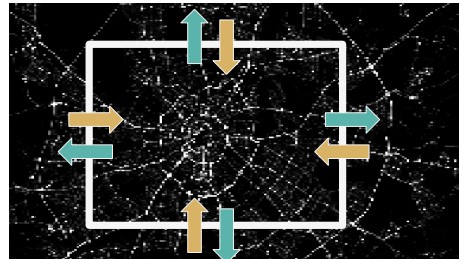

Figure 3: Schematic depiction of inbound-outbound traffic situations that require the conservation-of-vehicles principle. All vehicles on outbound roads (yellow arrows) must have entered the city center before (green arrows) or have been present in the first timestep.

## 5.3 PENDULUM WITH FRICTION

In the area of physics, we examined the usability of MC-LSTM for the problem of modeling a swinging pendulum with friction. Here, the total energy is the conserved property. During the movement of the pendulum, kinetic energy is converted into potential energy and vice-versa. This conversion between both energies has to be learned by the off-diagonal values of the redistribution matrix. A qualitative analysis of a trained MC-LSTM for this problem can be found in Appendix B.3.1.

Accounting for friction, energy dissipates and the swinging slows over time until towards a fixed point. This type of behavior presents a difficulty for machine learning and is impossible for methods that assume the pendulum to be a closed system, such as Hamiltonian networks (Greydanus et al., 2019). We generated 120 datasets of timeseries data of a pendulum where we used multiple different settings for initial angle, length of the pendulum, and the amount of friction. We then selected LSTM and MC-LSTM models and compared them with respect to predictive MSE. For an example, see Fig. 4. Overall, MC-LSTM has significantly outperformed LSTM with a mean MSE of $0.01$ (standard deviation $0.04$) compared to $0.05$ (standard deviation $0.15$; with a $p$-value $6.9e{-}8$, Wilcoxon test).

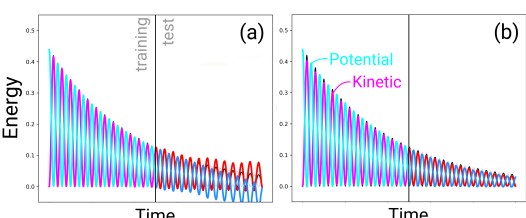

Figure 4: Example for the pendulum-modelling exercise. **(a)** LSTM trained for predicting energies of the pendulum with friction in auto-regressive fashion, **(b)** MC-LSTM trained in the same setting. Each subplot shows the potential- and kinetic energy and the respective predictions.

## 5.4 HYDROLOGY: RAINFALL RUNOFF MODELING

We tested MC-LSTM for large-sample hydrological modeling following Kratzert et al. (2019c). An ensemble of 10 MC-LSTMs was trained on 10 years of data from 447 basins using the publicly-available CAMELS dataset (Newman et al., 2015; Addor et al., 2017a). The mass input is precipitation and auxiliary inputs are: daily min. and max. temperature, solar radiation, and vapor pressure, plus 27 basin characteristics related to geology, vegetation, and climate (described by Kratzert et al., 2019c). All models besides MC-LSTM and LSTM were trained by different research groups with experience using each model. More details are given in Appendix B.4.2.

As shown in Tab. 3 MC-LSTM performed better with respect to the Nash–Sutcliffe Efficiency (NSE; the $R^2$ between simulated and observed runoff) than any other mass-conserving hydrology model, although slightly worse than LSTM.

NSE is often not the most important metric in hydrology, since water managers are typically concerned primarily with extremes (e.g. floods). MC-LSTM performed significantly better ($p = 0.025$, Wilcoxon test) than all models, including LSTM, with respect to high volume flows (FHV), at or above the 98th percentile flow in each basin. This makes MC-LSTM the current state-of-the-art model for flood prediction. MC-LSTM also performed significantly better than LSTM on low volume flows (FLV) and overall bias, however there are other hydrology models that are better for predicting low flows (which is important, e.g. for managing droughts).

Table 3: Hydrology benchmark results. All values represent the median (25% and 75% percentile in sub- and superscript, respectively) over the 447 basins. Only the two best performing hydrological models are included. An extended version can be found in Tab. B.7.

| Model | MC[a] | FHV[b] | NSE[c] |
|---|---|---|---|
| MC-LSTM | ✓ | **-14.7**$_{-23.4}^{-7.0}$ | $0.744_{0.641}^{0.814}$ |
| LSTM | ✗ | $-15.7_{-23.8}^{-8.6}$ | **0.763**$_{0.676}^{0.835}$ |
| mHM | ✓ | $-18.6_{-27.7}^{-9.5}$ | $0.666_{0.588}^{0.730}$ |
| ... | ... | ... | ... |
| HBVub | ✓ | $-18.5_{-27.8}^{-8.5}$ | $0.676_{0.578}^{0.749}$ |

[a]: *Mass conservation (MC).*
[b]: *Top 2% peak flow bias: $(-\infty, \infty)$, values closer to zero are desirable.*
[c]: *Nash-Sutcliffe Efficiency: $(-\infty, 1]$, values closer to one are desirable.*

**Model states and environmental processes.** It is an open challenge to bridge the gap between the fact that LSTM approaches give generally better predictions than other models (especially for flood prediction) and the fact that water managers need predictions that help them understand not only how much water will be in a river at a given time, but also how water moves through a basin.

Snow processes are difficult to observe and model. Kratzert et al. (2019a) showed that LSTM learns to track snow in memory cells without requiring snow data for training. We found similar behavior in MC-LSTMs, which has the advantage of doing this with memory cells that are *true* mass storages. Figure 5 shows the snow as the sum over a subset of MC-LSTM memory states and snow water equivalent (SWE) modeled by the well-established Snow-17 snow model (Anderson, 1973) (Pearson correlation coefficient $r \geq 0.91$). It is important to remember that MC-LSTMs did not have access to any snow data during training. In the best case it is possible to take advantage of the inductive bias to predict how much water will be stored as snow under different conditions by using simple combinations or mixtures of the internal states. Future work will determine whether this is possible with other difficult-to-observe states and fluxes.

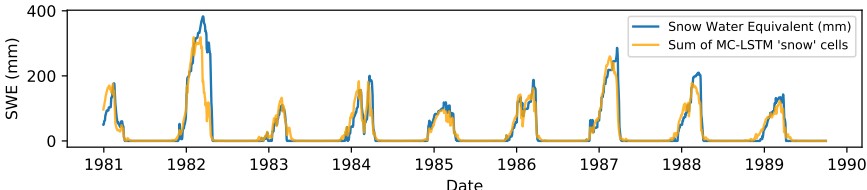

Figure 5: Snow-water-equivalent (SWE) from a single basin. The blue line is SWE modeled by Newman et al. (2015). The orange line is the sum over 4 MC-LSTM memory cells (Pearson correlation coefficient $r \geq 0.8$).

## 6    CONCLUSION.

We have demonstrated that with the concept of inductive biases an RNN can be designed that has the property to conserve mass of particular inputs. This architecture is highly proficient as neural arithmetic unit and is well-suited for predicting physical systems like hydrological processes, in which water mass has to be conserved.

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

## A    NOTATION OVERVIEW

Most of the notation used throughout the paper, is summarized in Tab. A.1.

Table A.1: Symbols and notations used in this paper.

| Definition | Symbol/Notation | Dimension |
|---|---|---|
| mass input at timestep $t$ | $\boldsymbol{x}^t$ or $x^t$ | $M$ or 1 |
| auxiliary input at timestep $t$ | $\boldsymbol{a}^t$ | $L$ |
| cell state at timestep $t$ | $\boldsymbol{c}^t$ | $K$ |
| limit of sequence of cell states | $\boldsymbol{c}^\infty$ | |
| hidden state at timestep $t$ | $\boldsymbol{h}^t$ | $K$ |
| redistribution matrix | $\boldsymbol{R}$ | $K \times K$ |
| input gate | $\boldsymbol{i}$ | $K$ |
| output gate | $\boldsymbol{o}$ | $K$ |
| mass | $\boldsymbol{m}$ | $K$ |
| input gate weight matrix | $\boldsymbol{W}_{\mathrm{i}}$ | $K \times L$ |
| input gate weight matrix | $\boldsymbol{W}_{\mathrm{o}}$ | $K \times L$ |
| output gate weight matrix | $\boldsymbol{U}_{\mathrm{i}}$ | $K \times K$ |
| output gate weight matrix | $\boldsymbol{U}_{\mathrm{o}}$ | $K \times K$ |
| identity matrix | $\boldsymbol{K}$ | $K \times K$ |
| input gate bias | $\boldsymbol{b}_{\mathrm{i}}$ | $K$ |
| output gate bias | $\boldsymbol{b}_{\mathrm{o}}$ | $K$ |
| arbitrary differentiable function | $f$ | |
| hypernetwork function (conditioner) | $g$ | |
| redistribution gate bias | $\boldsymbol{B}_{\mathrm{R}}$ | $K \times K$ |
| stored mass | $m_c$ | |
| mass efflux | $m_h$ | |
| limit of series of mass inputs | $m_x^\infty$ | |
| timestep index | $t$ | |
| an arbitrary timestep | $\tau$ | |
| last timestep of a sequence | $T$ | |
| redistribution gate weight tensor | $\mathsf{W}_{\mathrm{r}}$ | $K \times K \times L$ |
| redistribution gate weight tensor | $\mathsf{U}_{\mathrm{r}}$ | $K \times K \times K$ |
| arbitrary feature index | $a$ | |
| arbitrary feature index | $b$ | |
| arbitrary feature index | $c$ | |

## B    EXPERIMENTAL DETAILS

In the following, we provide further details on the experimental setups.

### B.1    NEURAL ARITHMETIC

Neural networks that learn arithmetic operations have recently come into focus (Trask et al., 2018; Madsen & Johansen, 2020). Specialized neural modules for arithmetic operations could play a role for complex AI systems since cognitive studies indicate that there is a part of the brain that enables animals and humans to perform basic arithmetic operations (Nieder, 2016; Gallistel, 2018). Although this primitive number processor can only perform approximate arithmetic, it is a fundamental part of our ability to understand and interpret numbers (Dehaene, 2011).

#### B.1.1    DETAILS ON DATASETS

We consider the *addition problem* that was proposed in the original LSTM paper (Hochreiter & Schmidhuber, 1997). We chose input values in the range $[0, 0.5]$ in order to be able to use the fast standard implementations of LSTM. For this task, 20 000 samples were generated using a fixed

random seed to create a dataset, which was split in 50% training and 50% validation samples. For the test data, a different random seed was used.

A definition of the *static arithmetic* task is provided by (Madsen & Johansen, 2020). The following presents this definition and its extension to the *recurrent arithmetic* task (c.f. Trask et al., 2018).

The input for the static version is a vector, $\boldsymbol{x} \in \mathcal{U}(1, 2)^{100}$, consisting of numbers that are drawn randomly from a uniform distribution. The target, $y$, is computed as

$$y = \left(\sum_{k=a}^{a+c} x_k\right) \square \left(\sum_{k=b}^{b+c} x_k\right),$$

where $c \in \mathbb{N}$, $a \leq b \leq a + c \in \mathbb{N}$ and $\square \in \{+, -, \cdot\}$. For the recurrent variant , the input consists of a sequence of $T$ vectors, denoted by $\boldsymbol{x}^t \in \mathcal{U}(1, 2)^{10}, t \in \{1, \ldots, T\}$, and the labels are computed as

$$y = \left(\sum_{t=1}^{T}\sum_{k=a}^{a+c} x_k^t\right) \square \left(\sum_{t=1}^{T}\sum_{k=b}^{b+c} x_k^t\right).$$

For these experiments, no fixed datasets were used. Instead, samples were generated on the fly. Note that since the subsets overlap, i.e., inputs are re-used, this data does not have mass conservation properties.

For a more detailed description of the *MNIST addition* data, we refer to (Trask et al., 2018) and the appendix of (Madsen & Johansen, 2020).

### B.1.2 DETAILS ON HYPERPARAMETERS.

For the *addition problem*, every network had a single hidden layer with 10 units. The output layer was a linear, fully connected layer for all MC-LSTM and LSTM variants. The NAU (Madsen & Johansen, 2020) and NALU/NAC (Trask et al., 2018) networks used their corresponding output layer. Also, we used a more common $L_2$ regularization scheme with low regularization constant ($10^{-4}$) to keep the weights ternary for the NAU, rather than the strategy used in the reference implementation from Madsen & Johansen (2020). Optimization was done using Adam (Kingma & Jimmy, 2015) for all models. The initial learning rate was selected from $\{0.1, 0.05, 0.01, 0.005, 0.001\}$ on the validation data for each method individually. All methods were trained for 100 epochs.

The weight matrices of LSTM were initialized in a standard way, using orthogonal and identity matrices for the forward and recurrent weights, respectively. Biases were initialized to be zero, except for the bias in the forget gate, which was initialized to 3. This should benefit the gradient flow for the first updates. Similarly, MC-LSTM is initialized so that the redistribution matrix (cf. Eq. 7) is (close to) the identity matrix. Otherwise we used orthogonal initialization (Saxe et al., 2014). The bias for the output gate was initialized to -3. This stimulates the output gates to stay closed (keep mass in the system), which has a similar effect as setting the forget gate bias in LSTM. This practically holds for all subsequently described experiments.

For the *recurrent arithmetic tasks*, we tried to stay as close as possible to the setup that was used by Madsen & Johansen (2020). This means that all networks had again a single hidden layer. The NAU, Neural Multiplication Unit (NMU) and NALU networks all had two hidden units and, respectively, NAU, NMU and NALU output layers. The first, recurrent layer for the first two networks was a NAU and the NALU network used a recurrent NALU layer. For the exact initialization of NAU and NALU, we refer to (Madsen & Johansen, 2020).

The MC-LSTM models used a fully connected linear layer with $L_2$-regularization for projecting the hidden state to the output prediction for the addition and subtraction tasks. It is important to use a free linear layer in order to compensate for the fact that the data does not have mass-conserving properties. However, it is important to note that the mass conservation in MC-LSTM is still necessary to solve this task. For the multiplication problem, we used a multiplicative, non-recurrent variant of MC-LSTM with an extra scalar parameter to allow the conserved mass to be re-scaled if necessary. This multiplicative layer is described in more detail in Appendix B.1.3.

Whereas the addition could be solved with two hidden units, MC-LSTM needed three hidden units to solve both subtraction and multiplication. This extra unit, which we refer to as the *trash cell*, allows

MC-LSTMs to get rid of excessive mass that should not influence the prediction. Note that, since the mass inputs are vectors, the input gate has to be computed in a similar fashion as the redistribution matrix. Adam was again used for the optimization. We used the same learning rate as Madsen & Johansen (2020) (0.001) to train the NAU, NMU and NALU networks. For MC-LSTM the learning rate was increased to 0.01 for addition and subtraction and 0.05 for multiplication after a manual search on the validation set. All models were trained for two million update steps.

In a similar fashion, we used the same models from (Madsen & Johansen, 2020) for the *MNIST addition* task. For MC-LSTM, we replaced the recurrent NAU layer with a MC-LSTM layer and the output layer was replaced with a fully connected linear layer. In this scenario, increasing the learning rate was not necessary. This can probably be explained by the fact that training CNN to regress the MNIST images is the main challenge during learning. We also used a standard $L_2$-regularization on the outputs of CNN instead of the implementation proposed in (Madsen & Johansen, 2020) for this task.

### B.1.3 STATIC ARITHMETIC

This experiment should enable a more direct comparison to the results from Madsen & Johansen (2020) than the recurrent variant. The data for the static task is equivalent to that of the recurrent task with sequence length one. For more details on the data, we refer to Appendix B.1.1 or (Madsen & Johansen, 2020).

Since the static task does not require a recurrent model, we discarded the redistribution matrix in MC-LSTM. The result is a layer with only input and output gates, which we refer to as an Mass-Conserving Fully Connected (MC-FC) layer. We compared this model to the results reported in (Madsen & Johansen, 2020), using the code base that accompanied the paper. All NALU and NAU networks had a single hidden layer. Similar to the recurrent task, MC-LSTM required two hidden units for addition and three for subtraction. Mathematically, an MC-FC with $K$ hidden neurons and $M$ inputs can be defined as MC-FC : $\mathbb{R}^M \to \mathbb{R}^K : \boldsymbol{x} \mapsto \boldsymbol{y}$, where

$$\boldsymbol{y} = \mathrm{diag}(\boldsymbol{o}) \cdot \boldsymbol{I} \cdot \boldsymbol{x} \qquad \boldsymbol{I} = \mathrm{softmax}(\boldsymbol{B}_I) \qquad \boldsymbol{o} = \sigma(\boldsymbol{b}_o),$$

where the softmax operates on the row dimension to get a column-normalized matrix, $\boldsymbol{I}$, for the input gate.

Using the log-exp transform (c.f. Trask et al., 2018), a multiplicative MC-FC with scaling parameter, $\boldsymbol{\alpha}$, can be constructed as follows, $\exp(\text{MC-FC}(\log(\boldsymbol{x})) + \boldsymbol{\alpha})$. The scaling parameter is necessary to break the mass conservation when it is not needed. By replacing the output layer with this multiplicative MC-FC, it can also be used to solve the multiplication problem. This network also required three hidden neurons. This model was compared to a NMU network with two hidden neurons and NALU network.

All models were trained for two million updates with the Adam optimizer (Kingma & Jimmy, 2015). The learning rate was set to 0.001 for all networks, except for the MC-FC network, which needed a lower learning rate of 0.0001, and the multiplicative MC-FC variant, which was trained with learning rate 0.01. These hyperparameters were found using a manual search.

Since the input consists of a vector, the input gate predicts a left-stochastic matrix, similar to the redistribution matrix. This allows us to verify generalization abilities of the inductive bias in MC-LSTMs. The performance was measured in a similar way as the recurrent task, except that generalization was tested over the range of the input values (Madsen & Johansen, 2020). Concretely, the models were trained on input values in $[1, 2]$ and tested on input values in the range $[2, 6]$. Table B.2 shows that MC-FC is able to match or outperform both NALU and NAU on this task.

### B.1.4 COMPARISON WITH TIME-DEPENDENT MC-LSTM

We used MC-LSTM with a time-independent redistribution matrix, as in Eq. (7), to solve the addition problem. This resembles another form of inductive bias, since we know that no redistribution across cells is necessary to solve this problem and it results also in a more efficient model, because less parameters have to be learned. However, for the sake of flexibility, we also verified that it is possible to use the more general time-dependent redistribution matrix (cf. Eq. 8). The results of this experiment can be found in Table B.3.

Table B.2: Results for the static arithmetic task. MC-FC is a mass-conserving variant of MC-LSTM based on fully-connected layers for non-recurrent tasks. MC-FCs for addition and subtraction/multiplication have two and three neurons, respectively. Error bars represent 95% confidence intervals.

| | addition | | subtraction | | multiplication | |
|---|---|---|---|---|---|---|
| | success rate[a] | updates[b] | success rate[a] | updates[b] | success rate[a] | updates[b] |
| MC-FC | $\mathbf{100\%} \, ^{+0\%}_{-4\%}$ | $2.1 \cdot 10^5$ | $\mathbf{100\%} \, ^{+0\%}_{-4\%}$ | $1.6 \cdot 10^5$ | $\mathbf{100\%} \, ^{+0\%}_{-4\%}$ | $1.4 \cdot 10^6$ |
| NAU / NMU | $\mathbf{100\%} \, ^{+0\%}_{-4\%}$ | $1.8 \cdot 10^4$ | $\mathbf{100\%} \, ^{+0\%}_{-4\%}$ | $5.0 \cdot 10^3$ | $98\% \, ^{+1\%}_{-5\%}$ | $1.4 \cdot 10^6$ |
| NAC | $\mathbf{100\%} \, ^{+0\%}_{-4\%}$ | $2.5 \cdot 10^5$ | $\mathbf{100\%} \, ^{+0\%}_{-4\%}$ | $9.0 \cdot 10^3$ | $31\% \, ^{+10\%}_{-8\%}$ | $2.8 \cdot 10^6$ |
| NALU | $14\% \, ^{+8\%}_{-5\%}$ | $1.5 \cdot 10^6$ | $14\% \, ^{+8\%}_{-5\%}$ | $1.9 \cdot 10^6$ | $0\% \, ^{+4\%}_{-0\%}$ | – |

[a] Percentage of runs that generalized to a different input range.
[b] Median number of updates necessary to solve the task.

Table B.3: Performance of different models on the LSTM addition task in terms of the MSE. MC-LSTM significantly (all $p$-values below .05) outperforms its competitors, LSTM (with high initial forget gate bias), NALU and NAU. Error bars represent 95%-confidence intervals across 100 runs.

| | reference[a] | seq length[b] | input range[c] | count[d] | combo[e] | NaN[f] |
|---|---|---|---|---|---|---|
| MC-LSTM[†] | $0.013 \pm 0.004$ | $0.022 \pm 0.010$ | $2.6 \pm 0.8$ | $2.2 \pm 0.7$ | $13.6 \pm 4.0$ | 0 |
| MC-LSTM | $\mathbf{0.004} \pm 0.003$ | $\mathbf{0.009} \pm 0.004$ | $\mathbf{0.8} \pm 0.5$ | $\mathbf{0.6} \pm 0.4$ | $\mathbf{4.0} \pm 2.5$ | 0 |
| LSTM | $0.008 \pm 0.003$ | $0.727 \pm 0.169$ | $21.4 \pm 0.6$ | $9.5 \pm 0.6$ | $54.6 \pm 1.0$ | 0 |
| NALU | $0.060 \pm 0.008$ | $0.059 \pm 0.009$ | $25.3 \pm 0.2$ | $7.4 \pm 0.1$ | $63.7 \pm 0.6$ | 93 |
| NAU | $0.248 \pm 0.019$ | $0.252 \pm 0.020$ | $28.3 \pm 0.5$ | $9.1 \pm 0.2$ | $68.5 \pm 0.8$ | 24 |

[a] training regime:     summing 2 out of 100 numbers between 0 and 0.5.
[b] longer sequence lengths:     summing 2 out of 1 000 numbers between 0 and 0.5.
[c] more *mass* in the input:     summing 2 out of 100 numbers between 0 and 5.0.
[d] higher number of summands:     summing 20 out of 100 numbers between 0 and 0.5.
[e] combination of previous scenarios:     summing 10 out of 500 numbers between 0 and 2.5.
[f] Number of runs that did not converge.
[†] MC-LSTM with time-dependent redistribution matrix.

Although the performance of MC-LSTM with time-dependent redistribution matrix is slightly worse than that of the more efficient MC-LSTM variant, it still outperforms all other models on the generalisation tasks. This can partly be explained by the fact that is harder to train a time-dependent redistribution matrix, and the training budget is limited to 100 epochs.

### B.1.5 QUALITATIVE ANALYSIS OF THE MC-LSTM MODELS TRAINED ON ARITHMETIC TASKS

**Addition Problem.** To reiterate, we used MC-LSTM with 10 hidden units and linear output layer. The model has to learn to sum all mass inputs of the timesteps, where the auxiliary input (the *marker*) equals $a^t = 1$, and ignore all other values. At the final timestep — where the auxiliary input equals $a^t = -1$ — the network should output the sum of all previously mark mass inputs.

In our experiment, the model has learned to store the marked input values in a single cell, while all other mass inputs mainly end up in a single, different cell. That is, a single cell learns to accumulate the inputs to compute the solution and the other cells are used as *trash cells*. In Fig. B.1, we visualize the cell states for a single input sample over time, where the orange and the blue line denote the mass accumulator and the main trash cell, respectively.

We can see that at the last time step — where the network is queried to return the accumulated sum — the value of that particular cell does not drop to zero (i.e., not the entire value that is actually accumulated is removed from the system). For this particular model, the corresponding output gate

value for this cell at the last time step is 0.18134. That is, only 18.134% of the actual accumulated value is returned. However, the weight of the linear layer that corresponds to this cell for this model is 5.5263. If we multiply these two values, the result is 1.0021, which means the model recovers the value stored in the cell state. For all other cells (grey lines), either the output gate at the last time step, the weight of the linear layer, or the cell value itself is zero. That means that the model output is only determined by the value of the single cell that acted as accumulator of the marked values (orange line).

We also analyzed MC-LSTM without linear output layer for the same addition task. In this case, the model output is determined as the sum over the outgoing mass. As before, the model also uses a single cell to store the accumulated values of the marked timesteps. However, because no scaling can be learned from the linear output layer, the model learned to fully *open* the output at the query timestep.

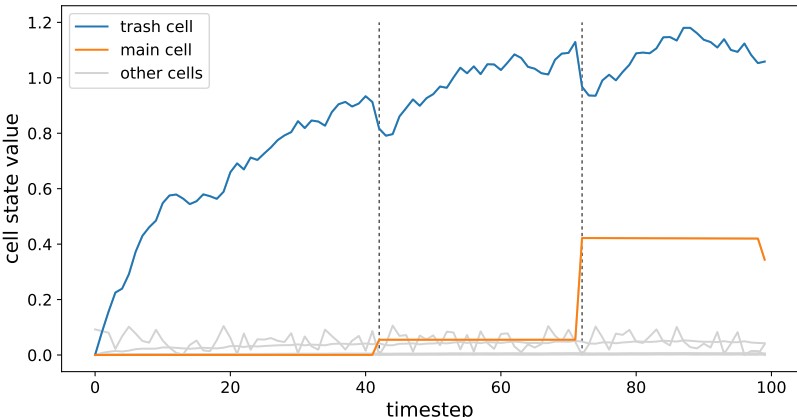

Figure B.1: MC-LSTM cell states over time for model trained to solve the *addition problem* (see Appendix B.1.1). Each line denotes the value of one particular cell over time, while the two vertical grey indicator lines denote the timesteps, where the auxiliary input was 1 (i.e., those numbers have to be added).

**Recurrent Arithmetic.**  In the following we take a closer look at the solution that is learned with MC-LSTM. Concretely, we look at the weights of a MC-LSTM model that successfully solves the following recurrent arithmetic task:

$$y = \sum_{t=1}^{T}(x_6^t + x_7^t) \, \Box \, \sum_{t=1}^{T}(x_7^t + x_8^t),$$

where $\Box \in \{-, +\}$, given a sequence of input vectors $x^t \in \mathbb{R}^{10}$ (the only purpose of the colors is to provide an aid to readers). We highlight the following observations:

1. For the addition task (i.e., $\Box \equiv +$), MC-LSTM has two units (see Appendix B.1.2 for details on the experiments). Trask et al. (2018); Madsen & Johansen (2020) fixed the number of hidden units to two with the idea that each unit can learn one term of the addition operation ($\Box$). However, if we take a look at the input gate of our model, we find that the first cell is used to accumulate $(x_1^t + \ldots + x_5^t + 0.5x_6^t + 0.5x_8^t + x_9^t + x_{10}^t)$ and the second cell collects $(0.5x_6^t + x_7^t + 0.5x_8^t)$. Since the learned redistribution matrix is the identity matrix, these accumulators operate individually.

   This means that, instead of computing the individual terms, MC-LSTM directly computes the solution, scaled by a factor ½ in its second cell. The first cell accumulates the rest of the mass, which it does not need for the prediction. In other words, it operates as some sort of *trash cell*. Note that due to the mass-conservation property, it would be impossible to

compute each side of the operation individually. After all, $x_7^t$ appears on both sides of the central operation ($\square$), and therefore the data is not mass conserving.

The output gate is always open for the *trash cell* and closed for the other cell, indicating that redundant mass is discarded through the output of the MC-LSTM in every timestep and the scaled solution is properly accumulated. However, in the final timestep — when the prediction is to be made, the output gate for the trash cell is closed and opened for the other cell. That is, the accumulated solution is passed to the final linear layer, which scales the output of MC-LSTM by a factor of two to get the correct solution.

2. For the subtraction task (i.e., $\square \equiv -$), a similar behavior can be observed. In this case, the final model requires three units to properly generalize. The first two cells accumulate $x_6^t$ and $x_8^t$, respectively. The last cell operates as *trash cell* and collects $(x_1^t + \ldots + x_5^t + x_7^t + x_9^t + x_{10}^t)$. The redistribution matrix is the identity matrix for the first two cells. For the *trash cell*, equal parts (0.4938) are redistributed to the two other cells. The output gate operates in a similar fashion as for addition. Finally, the linear layer computes the difference between the first two cells with weights 1, -1 and the *trash cell* is ignored with weight 0.

Although MC-LSTM with two units was not able to generalize well enough for the Madsen & Johansen (2020) benchmarks, it did turn out to be able to provide a reasonable solution (albeit with numerical flaws). With two cells, the network learned to store $(0.5x_1^t + \ldots + 0.5x_5^t + x_6^t + 0.5x_7^t + 0.5x_9^t + 0.5x_{10}^t)$ in one cell, and $(0.5x_1^t + \ldots + 0.5x_5^t + 0.5x_7^t + x_8^t + 0.5x_9^t + 0.5x_{10}^t)$ in the other cell. With a similar linear layer as for the three-unit variant, this solution should also compute a correct solution for the subtraction task.

## B.2 Inbound-outbound traffic forecast

Traffic forecasting considers a large number of different settings and tasks (Tedjopurnomo et al., 2020). For example whether the physical network topology of streets can be exploited by using graph neural networks combined with LSTMs (Cui et al., 2019). Within traffic forecasting mass conservation translates to a *conservation-of-vehicles* principle. Generally, models that adhere to this principle are desired (Vanajakshi & Rilett, 2004; Zhao et al., 2017) since they could be useful for long-term forecasts. Many recent benchmarking datasets for traffic forecasts are usually uni-directional and are measured at few streets. Thus conservation laws cannot be directly applied (Tedjopurnomo et al., 2020).

We demonstrate how MC-LSTM can be used in traffic forecasting settings. A typical setting for vehicle conservation is when traffic counts for inbound and outbound roads of a city are available. In this case, all vehicles that come from an inbound road must either be within a city or leave the city on an outbound road. The setting is similar to passenger flows in inbound and outbound metro (Liu et al., 2019), where LSTMs have also prevailed. We were able to extract such data from a recent dataset based on GPS-locations (Kreil et al., 2020) of vehicles at a fine geographic grid around cities, which represents good approximation of a vehicle conserving scenario.

**An approximately mass-conserving traffic dataset** Based on the data for the traffic4cast 2020 challenge (Kreil et al., 2020), we constructed a dataset to model inbound and outbound traffic of three different cities: Berlin, Istanbul and Moscow. The original data consists of 181 sequences of multi-channel images encoding traffic volume and speed for every five minutes in four (binned) directions. Every sequence corresponds to a single day in the first half of the year. In order to get the traffic flow from the multi-channel images at every timestep, we defined a frame around the city and collected the traffic-volume data for every pixel on the border of this frame. This is illustrated in Fig. 3. For simplicity, we ignored the fact that a single-pixel frame might have issues with fast-moving vehicles.

By taking into account the direction of the vehicles, the inbound and outbound traffic can be combined for every pixel on the border of our frame. To get a more tractable dataset, we additionally combined the pixels of the four edges of the frame to end up with eight values: four values for the incoming traffic, i.e: one for each border of the frame, and four values for the outgoing traffic. The inbound traffic would be the *mass* input for MC-LSTM and the target outputs are the outbound traffic along the different borders. The auxiliary input is the current daytime, encoded as a value between zero and one.

Table B.4: The hyperparameters resulting from the grid search for the traffic forecast experiment.

|  |  | hidden | lr | forget bias | initial state | learnable state |
|---|---|---|---|---|---|---|
| Berlin | LSTM | 10 | 0.01 | 0 | – | – |
|  | MC-LSTM | 100 | 0.01 | – | 0 | True |
| Istanbul | LSTM | 100 | 0.005 | 5 | – | – |
|  | MC-LSTM | 50 | 0.01 | – | 0 | False |
| Moscow | LSTM | 50 | 0.001 | 5 | – | – |
|  | MC-LSTM | 10 | 0.01 | – | 0 | False |

To model the sparsity that is often available in other traffic counting problems, we chose three time-slots (6 am, 12 pm and 6 pm) for which we use fifteen minutes of the actual measurements — i.e., three timesteps. This could for example simulate the deployment of mobile traffic counting stations. The other inputs are imputed by the average inbound traffic over the training data, which consists of 181 days. Outputs are only available when the actual measurements are used. This gives a total of 9 timesteps per day on which the loss can be computed. For training, this dataset is randomly split in 85% training and 15% validation samples.

During inference, all 288 timesteps of the inbound and outbound measurements are used to find out which model learned the traffic dynamics from the sparse training data best. For this purpose, we used the 18 days of validation data from the original dataset as test set, which are distributed across the second half of the year. In order to enable a fair comparison between LSTM and MC-LSTM, the data for LSTM was normalized to zero mean and unit variance for training and inference (using statistics from the training data). MC-LSTM does not need this pre-processing step and is fed the raw data.

**Model and Hyperparameters** For the traffic prediction, we used LSTM followed by a fully connected layer as baseline (c.f. Zhao et al., 2017; Liu et al., 2019). For MC-LSTM, we chose to enforce end-to-end mass conservation by using a MC-FC output layer, which is described in detail in Appendix B.1.3. For the initialization of the models, we refer to the details of the arithmetic experiments in Appendix B.1.

For each model and for each city, the best hyperparameters were found by performing a grid search on the validation data. This means that the hyperparameters were chosen to minimize the error on the nine 5-minute intervals. For all models, the number of hidden neurons was chosen from $\{10, 50, 100\}$ and for the learning rate, the options were $\{0.100, 0.050, 0.010, 0.005, 0.001\}$. All models were trained for $2\,000$ epochs using the Adam optimizer (Kingma & Jimmy, 2015). Additionally, we considered values in $\{0, 5\}$ for the initial value for the forget gate bias in LSTM. For MC-LSTM, the extra hyperparameters were the initial cell state value ($\in \{0, 100\}$) — i.e., how much cars are in each memory cell at timestep zero — and whether or not the initial cell state should be trained via backpropagation. The results of the hyperparameter search can be found in Tab. B.4.

The idea behind tuning the initial cell state, is that unlike with LSTM, the cell state in MC-LSTM directly reflects the number of cars that can drive out of a city during the first timesteps. If the initial cell state is too high or too low, this might negatively affect the prediction capabilities of the model. If it would be possible to estimate the number of cars in a city at the start of the sequence, this could also be used to get better estimates for the initial cell state. However, from the results of the hyperparameter search (see Tab. B.4), we might have overestimated the importance of these hyperparameters.

**Results.** All models were evaluated on the test data, using the checkpoint after $2\,000$ epochs for fifty runs. An example of what the predictions of both models look like for an arbitrary day in an arbitrarily chosen city is displayed in Fig. B.2. The average Root MSE (RMSE) and Mean Absolute Error (MAE) are summarized in Tab. B.5. The results show that MC-LSTM is able to generalize significantly better than LSTM for this task. The RMSE of MC-LSTM is significantly better than

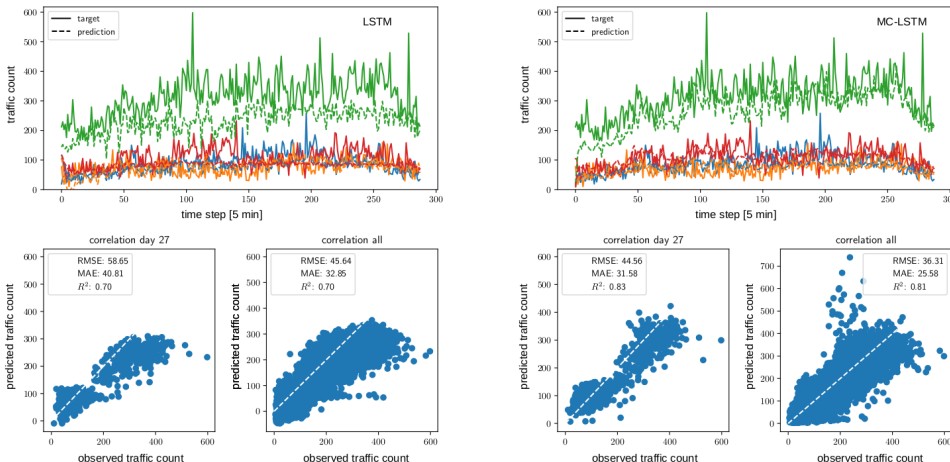

Figure B.2: Traffic forecasting models for outbound traffic in Moscow. An arbitrary day has been chosen for display. Note that both models have only been trained on data at timesteps 71-73, 143-145, and 215-217. Colors indicate the four borders of the frame, i.e. north, east, south and west. **Left:** LSTM predictions shown in dashed lines versus the actual traffic counts (solid lines). **Right:** MC-LSTM predictions shown in dashed lines versus the actual traffic counts (solid lines).

Table B.5: Results on outbound traffic forecast avg RMSE and MAE with 95% confidence intervals over 50 runs

|  | Istanbul | | Berlin | | Moscow | |
|---|---|---|---|---|---|---|
|  | RMSE | MAE | RMSE | MAE | RMSE | MAE |
| MC-LSTM | **7.3** ± 0.1 | **28** ± 2 | **13.6** ± 1.8 | **66** ± 1 | 25.5 ± 1.1 | **27.8** ± 1.1 |
| LSTM | 142.6 ± 4.4 | 84 ± 3 | 135.4 ± 5.0 | 84 ± 3 | 45.6 ± 0.8 | 31.7 ± 0.5 |

LSTM ($p$-values $4e-10$, $8e-3$, and $4e-10$ for Istanbul, Berlin, and Moscow, respectively, Wilcoxon test).

### B.3 PENDULUM WITH FRICTION

In the area of physics, we consider the problem of modeling a swinging pendulum with friction. The conserved quantity of interest is the total energy. During the movement of the pendulum, kinetic energy is converted into potential energy and vice-versa. Neglecting friction, the total energy is conserved and the movement would continue indefinitely. Accounting for friction, energy dissipates and the swinging slows over time until a fixed point is reached. This type of behavior presents a difficulty for machine learning and is impossible for methods that assume the pendulum to be closed systems, such as Hamiltonian networks (Greydanus et al., 2019). We postulated that both energy conversion and dissipation can be fitted by machine learning models, but that an appropriate inductive bias will allow to generalize from the learned data with more ease.

To train the model, we generated a set of timeseries using the differential equations for a pendulum with friction. We used multiple different settings for initial angle, length of the pendulum, the amount of friction, the length of the training-period and with and without Gaussian noise. Each model received the initial kinetic and potential energy of the pendulum and must predict the consecutive timesteps. The time series starts always with the pendulum at the maximum displacement — i.e., the entire energy in the system is potential energy. We generated timeseries of potential- and kinetic energies by iterating the following settings/conditions: initial amplitude ($\{0.2, 0.4\}$), pendulum length ($\{0.75, 1\}$), length of training sequence in terms of timesteps ($\{100, 200, 400\}$), noise level ($\{0, 0.01\}$), and dampening constant ($\{0.0, 0.1, 0.2, 0.4, 0.8\}$). All combinations of those settings

were used to generate a total of 120 datasets, for which we train both models (the auto-regressive LSTM and MC-LSTM).

We trained an auto-regressive LSTM that receives its current state and a low-dimensional temporal embedding (using nine sinusoidal curves with different frequencies) to predict the potential and kinetic energy of the pendulum. Similarly, MC-LSTM is trained in an autoregressive mode, where a hypernetwork obtains the current state and the same temporal embedding as LSTM. The model-setup is thus similar to an autoregressive model with exogenous variables from classical timeseries modelling literature. To obtain suitable hyperparameters we manualy adjusted the learning rate (0.01), hidden size of LSTM (256), the hypernetwork for estimating the redistribution (a fully connected network with 3 layers, ReLu activations and hidden sizes of 50, 100, and 2 respectively), optimizer (Adam, Kingma & Jimmy, 2015) and the training procedure (crucially, the amount of additionally considered timesteps in the loss after a threshold is reached. See explanation of the used loss below), on a separately generated validation dataset.

For MC-LSTM, a hidden size of two was used so that each state directly maps to the two energies. The hypernetwork consists of three fully connected layers of size 50, 100 and 4, respectively. To account for the critical values at the extreme-points of the pendulum (i.e. the amplitudes — where the energy is present only in the form of potential energy — and the midpoint — where only kinetic energy exists), we slightly offset the cell state from the actual predicted value by using a linear regression with a slope of $1.02$ and an intercept $-0.01$.

For both models, we used a combination of Pearson's correlation of the energy signals and the MSE as a loss function (by subtracting the former mean from the latter). Further, we used a simple curriculum to deal with the long autoregressive nature of the timeseries (Bengio et al., 2015): Starting at a time window of eleven we added five additional timesteps whenever the combined loss was below $-0.9$.

Overall, MC-LSTM has significantly outperformed LSTM with a mean MSE of $0.01$ (standard deviation $0.04$) compared to $0.05$ (standard deviation $0.15$; with a $p$-value $6.9\mathrm{e}{-8}$, Wilcoxon test).

### B.3.1 QUALITATIVE ANALYSIS OF THE MC-LSTM MODELS TRAINED FOR A PENDULUM

In the following, we analyse the behavior of the simplest pendulum setup, i.e., the one without friction. Special to the problem of the pendulum without friction is that there are no mass in- or outputs and the whole dynamic of the system has to be modeled by the redistribution matrix. The initial state of the system is given by the displacement of the pendulum at the start, where all energy is stored as potential energy. Afterwards, the pendulum oscillates, converting potential to kinetic energy and vice-versa.

In MC-LSTM, the conversion between the two forms of energy has to be learned by the redistribution matrix. More specifically, the off-diagonal elements denote the fraction of energy that is converted from one form to the other. In contrast, the diagonal elements of the redistribution matrix denote the fraction of energy that is *not* converted.

In Fig. B.3, we visualize the off-diagonal elements of the redistribution matrix (i.e., the conversion of energy) for the pendulum task without friction, as well as the modeled potential and kinetic energy. We can see that an increasing fraction of energy is converted into the other form, until the total energy of the system is stored as either kinetic or potential energy. As soon as the total energy is e.g. converted into kinetic energy, the corresponding off-diagonal element (the orange line of the upper plot in Fig. B.3) drops to zero. Here, the other off-diagonal element (the blue line of the upper plot in Fig. B.3) starts to increase, meaning that energy is converted back from kinetic into potential energy. Note that the differences in the maximum values of the off-diagonal elements is not important, since at this point the corresponding energy is already approximately zero.

### B.4 HYDROLOGY

Modeling river discharge from meteorological data (e.g., precipitation, temperature) is one of the most important tasks in hydrology, and is necessary for water resource management and risk mitigation related to flooding. Recently, Kratzert et al. (2019c; 2020) established LSTM-based models as state-of-the-art in rainfall runoff modeling, outperforming traditional hydrological models by a large margin against most metrics (including peak flows, which is critical for flood prediction). However, the hydrology community is still reluctant to adopt these methods (e.g. Beven, 2020). A recent

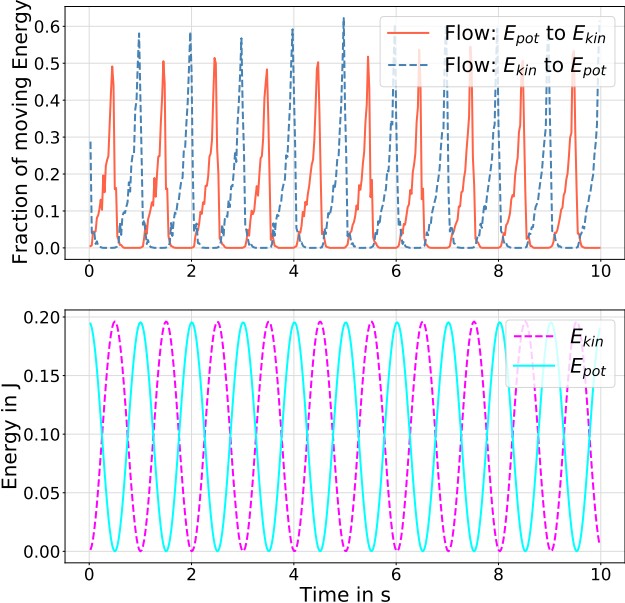

Figure B.3: Redistribution of energies in a pendulum learned by MC-LSTM. The upper plot shows the fraction of energy that is redistributed between the two cells that model $E_{pot}$ and $E_{kin}$ over time. The continuous redistribution of energy results in the two time series of potential and kinetic energy displayed in the lower plot.

workshop on 'Big Data and the Earth Sciences' Sellars (2018) reported that *"[m]any participants who have worked in modeling physical-based systems continue to raise caution about the lack of physical understanding of ML methods that rely on data-driven approaches."*

One of of the most basic principles in watershed modeling is mass conservation. Whether water is treated as a resource (e.g. droughts) or hazard (e.g. floods), a modeller must be sure that they are accounting for all of the water in a catchment. Thus, most models conserve mass (Todini, 1988), and attempt to explicitly implement the most important physical processes. The downside of this 'model everything' strategy is that errors are introduced for every real-world process that is *not* implemented in a model, or implemented incorrectly. In contrast, MC-LSTM is able to learn any necessary behavior that can be induced from the signal (like LSTM) while still conserving the overall water budget.

### B.4.1 DETAILS ON THE DATASET

The data used in all hydrology related experiments is the publicly available Catchment Attributes and Meteorology for Large-sample Studies (CAMELS) dataset (Newman et al., 2014; Addor et al., 2017b). CAMELS contains data for 671 basins and is curated by the US National Center for Atmospheric Research (NCAR). It contains only basins with relatively low anthropogenic influence (e.g., dams and reservoirs) and basin sizes range from 4 to 25 000 km$^2$. The basins cover a range of different geo- and eco-climatologies, as described by Newman et al. (2015) and Addor et al. (2017a). Out of all 671 basins, we used 447 — these are the basins for which simulations from all benchmark models are available (see Sec. B.4.4). To reiterate, we used benchmark hydrology models that were trained and tested by other groups with experience using these models, and were therefore limited to the 447 basis with results for all benchmark models. The spatial distribution of the 447 basins across the contiguous USA (CONUS) is shown in Fig. B.4.

For each catchment, roughly 30 years of daily meteorological data from three different products exist (DayMet, Maurer, NLDAS). Each meteorological dataset consist of five different variables: daily cumulative precipitation, daily minimum and maximum temperature, average short-wave radiation and vapor pressure. We used the Maurer forcing data because this is the data product that was used by all benchmark models (see Sec. B.4.4). In addition to meteorological data, CAMELS also

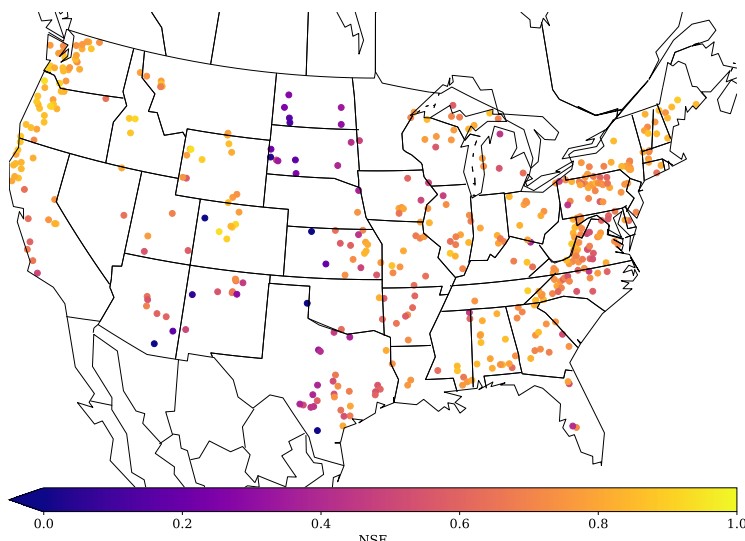

Figure B.4: Spatial distribution of the 447 catchments considered in this study. The color denotes the Nash-Sutcliffe Efficiency of the MC-LSTM ensemble for each basin, where a value of 1 means perfect predictions.

includes a set of static catchment attributes derived from remote sensing or CONUS-wide available data products. The static catchment attributes can broadly be grouped into climatic, vegetation or hydrological indices, as well as soil and topological properties. In this study, we used the same 27 catchment attributes as Kratzert et al. (2019c). Target data were daily averaged streamflow observations originally from the USGS streamflow gauge network, which are also included in the CAMELS dataset.

**Training, validation and test set.** Following the calibration and test procedure of the benchmark hydrology models, we trained on streamflow observations from 1 October 1999 through 30 September 2008 and tested on observations from 1 October 1989 to 30 September 1999. The remaining period (1 October 1980 to 30 September 1989) was used as validation period for hyperparameter tuning.

### B.4.2 DETAILS ON THE TRAINING SETUP AND MC-LSTM HYPERPARAMETERS

The general model setup follows insights from previous studies (Kratzert et al., 2018; 2019c;b; 2020), where LSTMs were used for the same task. We use sequences of 365 timesteps (days) of meteorological inputs to predict discharge at the last timestep of the sequence (sequence-to-one prediction). The mass input $x$ in this experiment was catchment averaged precipitation (mm/day) and the auxiliary inputs $a$ were the 4 remaining meteorological variables (min. and max. temperature, short-wave radiation and vapor pressure) as well as the 27 static catchment attributes, which are constant over time.

We tested a variety of MC-LSTM model configurations and adaptions for this specific task, which are briefly described below:

1. **Processing auxiliary inputs with LSTM**: Instead of directly using the auxiliary inputs in the input gate (Eq. 5), output gate (Eq. 6) and time-dependent mass redistribution (Eq. 8), we first processed the auxiliary inputs a with LSTM and then used the output of this LSTM as the auxiliary inputs. The idea was to add additional memory for the auxiliary inputs, since in its base form only mass can be stored in the cell states of MC-LSTM. This could be seen as a specific adaption for the rainfall runoff modeling application, since information about the weather today and in the past ought to be useful for controlling the gates and mass redistribution. Empirically however, we could not see any significant performance gain and therefore decided to not use the more complex version with an additional LSTM.

2. **Auxiliary output + regularization to account for evapotranspiration**: Of all precipitation falling in a catchment, only a part ends as discharge in the river. Large portions of precipitation are lost to the atmosphere in form of evaporation (from e.g. open water surfaces) and transpiration (from e.g. plants and trees), and to groundwater. One approach to account for this "mass loss" is the following: instead of summing over outgoing mass (Eq. 4), we used a linear layer to connect the outgoing mass to two output neurons. One neuron was fitted against the observed discharge data, while the second was used to estimate water loss due to unobserved sinks. A regularization term was added to the loss function to account for this. This regularization term was computed as the difference between the sum of the outgoing mass from MC-LSTM and the sum over the two output neurons. This did work, and the timeseries of the second auxiliary output neuron gave interesting results (i.e. matching the expected behavior of the annual evapotranspiration cycle), however results were not significantly better compared to our final model setup, which is why we rejected this architectural change.

3. **Explicit trash cell** Another way to account for evapotranspiration that we tested is to allow the model to use one memory cell as explicit "trash cell". That is, instead of deriving the final model prediction as the sum over the *entire* outgoing mass vector, we only calculate the sum over all but e.g. one element (see Eq. 13). This simple modification allows the model to use e.g. the first memory cell to discard mass from the system, which is then ignored for the model prediction. We found that this modification improved performance, and thus integrated it into our final model setup.

4. **Input/output scaling to account for input/output uncertainty**: Both, input and output data in our applications inherit large uncertainties (Nearing et al., 2016), which is not ideal for mass-conserving models (and likely one of the reasons why LSTM performs so well compared to all other mass-conserving models). To account for that, we tried three different adaptions. First, we used a small fully connected network to derive time-dependent scaling weights for the mass input, which we regularized to be close to one. Second, we used a linear layer with positive weights to map the outgoing mass to the final model prediction, where all weights were initialized to one and the bias to zero. Third, we combined both. Out of the three, the input scaling resulted in the best performing model, however the results were worse than not scaling.

5. **Time-dependent redistribution matrix variants**: For this experiment, a time-dependent redistribution matrix is necessary, since the underlying real-world processes (such as snow melt and thus conversion from snow into e.g. soil moisture or surface runoff) are time-dependent. Since using the redistribution matrix as proposed in Eq. 8 is memory-demanding, especially for models with larger numbers of memory cells, we also tried to use a different method for this experiment. Here, we learned a fixed matrix (as in Eq. 7) and only calculated two vectors for each timestep. The final redistribution matrix was then derived as the outer product of the two time-dependent vectors and the static matrix. This resulted in lower memory consumption, however the model performance deteriorated significantly, which could be a hint towards the complexity required to learn the redistributing processes in this problem.

6. **Activation function of the redistribution matrix**: We tested several different activation functions for the redistribution matrix in this experiment. Among those were the normalized sigmoid function (that is used e.g. for the input gate), the softmax function (as in Eq. 8) and the normalized ReLU activation function (see Eq. 18). We could achieve the best results using the normalized ReLU variant and can only hypothesize the reason for that: In this application (rainfall-runoff modelling) there are several state processes that are strictly disconnected. One example is snow and groundwater: groundwater will never turn into snow and snow will never transform into groundwater (not directly at least, it will first need to percolate through upper soil layers). Using normalized sigmoids or softmax makes it numerically harder (or impossible) to not distributed at least *some* mass between every cell — because activations can never be exactly zero. The normalized ReLU activation can do so, however, which might be the reason that it worked better in this case.

As an extension to the standard MC-LSTM model introduced in Eq. (5) to Eq. (8), we also used the mass input (precipitation) in all gates. The reason is the following: Different amounts of precipitations can lead to different processes. For example, low amounts of precipitation could be absorbed by the

soil and stored as soil moisture, leading to effectively no immediate discharge contribution. Large amounts of precipitation on the other hand, could lead to direct surface runoff, if the water cannot infiltrate the soil at the rate of the precipitation falling down. Therefore, it is crucial that the gates have access to the information contained in the precipitation input. The final model design used in all hydrology experiments is described by the following equations:

$$\boldsymbol{m}_{\text{tot}}^t = \boldsymbol{R}^t \cdot \boldsymbol{c}^{t-1} + \boldsymbol{i}^t \cdot x^t \tag{10}$$

$$\boldsymbol{c}^t = (\boldsymbol{1} - \boldsymbol{o}^t) \odot \boldsymbol{m}_{\text{tot}}^t \tag{11}$$

$$\boldsymbol{h}^t = \boldsymbol{o}^t \odot \boldsymbol{m}_{\text{tot}}^t \tag{12}$$

$$\widehat{y} = \sum_{i=2}^{n} h_i^t, \tag{13}$$

with the gates being defined by

$$\boldsymbol{i}^t = \tilde{\sigma}\left(\boldsymbol{W}_{\text{i}} \cdot \boldsymbol{a}^t + \boldsymbol{U}_{\text{i}} \cdot \frac{\boldsymbol{c}^{t-1}}{\|\boldsymbol{c}^{t-1}\|_1} + \boldsymbol{V}_{\text{i}} \cdot x^t + \boldsymbol{b}_{\text{i}}\right) \tag{14}$$

$$\boldsymbol{o}^t = \sigma\left(\boldsymbol{W}_{\text{o}} \cdot \boldsymbol{a}^t + \boldsymbol{U}_{\text{o}} \cdot \frac{\boldsymbol{c}^{t-1}}{\|\boldsymbol{c}^{t-1}\|_1} + \boldsymbol{V}_{\text{o}} \cdot x^t + \boldsymbol{b}_{\text{o}}\right) \tag{15}$$

$$\boldsymbol{R}^t = \widetilde{\text{ReLU}}\left(\mathbf{W}_{\text{r}} \cdot \boldsymbol{a}^t + \mathbf{U}_{\text{r}} \cdot \frac{\boldsymbol{c}^{t-1}}{\|\boldsymbol{c}^{t-1}\|_1} + \mathbf{V}_{\text{r}} \cdot x^t + \boldsymbol{B}_{\text{r}}\right), \tag{16}$$

where $\tilde{\sigma}$ is the *normalized logistic function* and $\widetilde{\text{ReLU}}$ is the *normalized rectified linear unit* (ReLU) that we define in the following. The normalized logistic function defined of the input gate is defined by:

$$\tilde{\sigma}(i_k) = \frac{\sigma(i_k)}{\sum_k \sigma(i_k)}. \tag{17}$$

In this experiment, the activation function for the redistribution gate is the normalized ReLU function defined by:

$$\widetilde{\text{ReLU}}(s_k) = \frac{\max(s_k, 0)}{\sum_k \max(s_k, 0)}, \tag{18}$$

where $\boldsymbol{s}$ is some input vector to the normalized ReLU function.

We manually tried different sets of hyperparameters, because a large-scale automatic hyperparameter search was not feasible. Besides trying out all variants as described above, the main hyperparameter that we tuned for the final model was the number of memory cells. For other parameters, such as learning rate, mini-batch size, number of training epochs, we relied on previous work using LSTMs on the same dataset.

The final hyperparameters are a hidden size of 64 memory cells and a mini-batch size of 256. We used the Adam optimizer (Kingma & Jimmy, 2015) with a scheduled learning rate starting at 0.01 then lowering the learning rate after 20 epochs to 0.005 and after another 5 epochs to 0.001. We trained the model for a total number of 30 epochs and used the weights of the last epoch for the final model evaluation. All weight matrices were initialized as (semi) orthogonal matrices (Saxe et al., 2014) and all bias terms with a constant value of zero. The only exception was the bias of the output gate, which we initialized to $-3$, to keep the output gate closed at the beginning of the training.

### B.4.3 DETAILS ON THE LSTM MODEL

For LSTM, we largely relied on expertise from previous studies (Kratzert et al., 2018; 2019c;b; 2020). The only hyperparameter we adapted was the number of memory cells, since we used fewer basins

Table B.6: Model robustness of MC-LSTM and LSTM results over the $n = 10$ different random seeds. For all $n = 10$ models, we calculated the median performance for each metric and report the mean and standard deviation of the median values in this table.

| | MC[a] | NSE[b] | $\beta$-NSE[c] | FLV[d] | FHV[e] |
|---|---|---|---|---|---|
| MC-LSTM Single | ✓ | 0.726±0.003 | -0.021±0.003 | -38.7±3.2 | -13.9±0.7 |
| LSTM Single | ✗ | 0.737±0.003 | -0.035±0.005 | 13.6±3.4 | -14.8±1.0 |

[a]: *Mass conservation (MC)*.
[b]: *Nash-Sutcliffe efficiency: $(-\infty, 1]$, values closer to one are desirable.*
[c]: *$\beta$-NSE decomposition: $(-\infty, \infty)$, values closer to zero are desirable.*
[d]: *Bottom 30% low flow bias: $(-\infty, \infty)$, values closer to zero are desirable.*
[e]: *Top 2% peak flow bias: $(-\infty, \infty)$, values closer to zero are desirable.*

(447) than in the previous studies (531). We found that LSTM with 128 memory cells, compared to the 256 used in previous studies, resulted in slightly better results. Apart from that, we trained LSTMs with the same inputs and settings (sequence-to-one with a sequence length of 365) as described in the previous section for MC-LSTM. We used the standard LSTM implementation from the PyTorch package (Paszke et al., 2019), i.e., with forget gate (Gers et al., 2000). We manually initialized the bias of the forget gate to be 3 in order to keep the forget gate open at the beginning of the training.

### B.4.4 DETAILS ON THE BENCHMARK MODELS

The benchmark models were first collected by Kratzert et al. (2019c). All models were configured, trained and run by several different research groups, most often the respective model developers themselves. This was done to avoid any potential to favor our own models. All models used the same forcing data (Maurer) and the same time periods to train and test. The models can be classified in two groups:

1. *Models trained for individual watersheds.* These are SAC-SMA (Newman et al., 2017), VIC (Newman et al., 2017), three different model structures of FUSE[1], mHM (Mizukami et al., 2019) and HBV (Seibert et al., 2018). For the HBV model, two different simulations exist: First, the ensemble average of 1000 untrained HBV models (lower benchmark) and second, the ensemble average of 100 trained HBV models (upper benchmarks). For details see (Seibert et al., 2018).

2. *Models trained regionally.* For hydrological models, regional training means that one parameter transfer model was trained, which estimates watershed-specific model parameters through globally trained model functions from e.g. soil maps or other catchment attributes. For this setting, the benchmark dataset includes simulations of the VIC model (Mizukami et al., 2017) and mHM (Rakovec et al., 2019).

### B.4.5 DETAILED RESULTS.

Table B.6 provides results for MC-LSTM and LSTM averaged over the $n = 10$ model repetitions.

Table B.7 provides the complete benchmarking results.

---

[1]Provided by Nans Addor on personal communication

Table B.7: Full hydrology benchmark results. All values represent the median (25% and 75% percentile in sub- and superscript, respectively) over the 447 basins. For both MC-LSTM and LSTM, metrics were derived from the ensemble mean prediction over the 10 model repetitions (same as reported in the main paper).

| | MC[a] | NSE[b] | $\beta$-NSE[c] | FLV[d] | FHV[e] |
|---|---|---|---|---|---|
| MC-LSTM Ensemble | ✓ | $0.744^{0.814}_{0.641}$ | $-0.020^{0.013}_{-0.066}$ | $-24.7^{31.1}_{-94.4}$ | $-14.7^{-7.0}_{-23.4}$ |
| LSTM Enssemble | ✗ | $0.763^{0.835}_{0.676}$ | $-0.034^{-0.002}_{-0.077}$ | $36.3^{59.7}_{-0.4}$ | $-15.7^{-8.6}_{-23.8}$ |
| SAC-SMA | ✓ | $0.603^{0.682}_{0.512}$ | $-0.066^{-0.026}_{-0.108}$ | $37.4^{68.1}_{-31.9}$ | $-20.4^{-12.2}_{-29.9}$ |
| VIC (basin) | ✓ | $0.551^{0.641}_{0.465}$ | $-0.018^{0.032}_{-0.071}$ | $-74.8^{23.1}_{-271.8}$ | $-28.1^{-17.5}_{-40.1}$ |
| VIC (regional) | ✓ | $0.307^{0.402}_{0.218}$ | $-0.074^{0.023}_{-0.166}$ | $18.9^{69.6}_{-73.1}$ | $-56.5^{-38.3}_{-64.6}$ |
| mHM (basin) | ✓ | $0.666^{0.730}_{0.588}$ | $-0.040^{0.003}_{-0.102}$ | $11.4^{65.1}_{-64.0}$ | $-18.6^{-9.5}_{-27.7}$ |
| mHM (regional) | ✓ | $0.527^{0.619}_{0.391}$ | $-0.039^{0.033}_{-0.169}$ | $36.8^{70.9}_{-32.6}$ | $-40.2^{-23.8}_{-51.0}$ |
| HBV (lower) | ✓ | $0.417^{0.550}_{0.276}$ | $-0.023^{0.058}_{-0.114}$ | $23.9^{61.0}_{-25.9}$ | $-41.9^{-17.3}_{-55.2}$ |
| HBV (upper) | ✓ | $0.676^{0.749}_{0.578}$ | $-0.012^{0.034}_{-0.058}$ | $18.3^{67.5}_{-62.9}$ | $-18.5^{-8.5}_{-27.8}$ |
| FUSE (900) | ✓ | $0.639^{0.715}_{0.539}$ | $-0.031^{0.024}_{-0.100}$ | $-10.5^{49.2}_{-94.8}$ | $-18.9^{-9.9}_{-27.8}$ |
| FUSE (902) | ✓ | $0.650^{0.727}_{0.570}$ | $-0.047^{-0.004}_{-0.098}$ | $-68.2^{17.1}_{-239.9}$ | $-19.4^{-8.9}_{-27.9}$ |
| FUSE (904) | ✓ | $0.622^{0.705}_{0.527}$ | $-0.067^{-0.019}_{-0.135}$ | $-67.6^{35.7}_{-238.6}$ | $-21.4^{-11.3}_{-33.0}$ |

[a]: *Mass conservation (MC).*
[b]: *Nash-Sutcliffe efficiency:* $(-\infty, 1]$*, values closer to one are desirable.*
[c]: *$\beta$-NSE decomposition:* $(-\infty, \infty)$*, values closer to zero are desirable.*
[d]: *Bottom 30% low flow bias:* $(-\infty, \infty)$*, values closer to zero are desirable.*
[e]: *Top 2% peak flow bias:* $(-\infty, \infty)$*, values closer to zero are desirable.*

## C   THEOREMS & PROOFS

**Theorem 1** (Conservation property). *Let $m_c^\tau = \sum_k c_k^\tau$ and $m_h^\tau = \sum_k h_k^\tau$ be, respectively, the mass in the MC-LSTM storage and the outputs at time $\tau$. At any timestep $\tau$, we have:*

$$m_c^\tau = m_c^0 + \sum_{t=1}^{\tau} x^t - \sum_{t=1}^{\tau} m_h^t.$$

*That is, the change of mass in the cell states is the difference between input and output mass, accumulated over time.*

*Proof.* The proof is by induction and we use $\boldsymbol{m}_{\text{tot}} = \boldsymbol{R}^t \cdot \boldsymbol{c}^{t-1} + \boldsymbol{i}^t \cdot x^t$ from Eq.(2).

For $\tau = 0$, we have $m_c^0 = m_c^0 + \sum_{t=1}^{0} x^t - \sum_{t=1}^{0} m_h^t$, which is trivially true when using the convention that $\sum_{t=1}^{0} = 0$.

Assuming that the statement holds for $\tau = T$, we show that it must also hold for $\tau = T + 1$.

Starting from Eq. (3), the mass of the cell states at time $T + 1$ is given by:

$$m_c^{T+1} = \sum_{k=1}^{K}(1 - o_k)m_{\text{tot},k}^{T+1} = \sum_{k=1}^{K} m_{\text{tot},k}^{T+1} - \sum_{k=1}^{K} o_k m_{\text{tot},k}^{T+1},$$

where $m_{\text{tot},k}^t$ is the $k$-th entry of the result from Eq. (2) (at timestep $t$). The sum over entries in the first term can be simplified as follows:

$$\sum_{k=1}^{K} m_{\text{tot},k}^{T+1} = \sum_{k=1}^{K} \left( \sum_{j=1}^{K} r_{kj} c_j^T + i_k x^{T+1} \right)$$
$$= \sum_{j=1}^{K} c_j^T \left( \sum_{k=1}^{K} r_{kj} \right) + x^{T+1} \sum_{k=1}^{K} i_k$$
$$= m_c^T + x^{T+1}.$$

The final simplification is possible because $\boldsymbol{R}$ and $\boldsymbol{i}$ are (left-)stochastic. The mass of the outputs can then be computed from Eq. (4):

$$m_h^{T+1} = \sum_{k=1}^{K} o_k m_{\text{tot},k}^{T+1}.$$

Putting everything together, we find

$$m_c^{T+1} = \sum_{k=1}^{K} m_{\text{tot},k}^{T+1} - \sum_{k=1}^{K} o_k m_{\text{tot},k}^{T+1}$$
$$= m_c^T + x^{T+1} - m_h^{T+1}$$
$$= m_c^0 + \sum_{t=1}^{T} x^t - \sum_{t=1}^{T} m_h^t + x^{T+1} - m_h^{T+1}$$
$$= m_c^0 + \sum_{t=1}^{T+1} x^t - \sum_{t=1}^{T+1} m_h^t$$

By the principle of induction, we conclude that mass is conserved, as specified in Eq. (9).   $\square$

**Corollary 1.** *In each timestep $\tau$, the cell states $c_k^\tau$ are bounded by the sum of mass inputs $\sum_{t=1}^{\tau} x^\tau + m_c^0$, that is $|c_k^\tau| \le \sum_{t=1}^{\tau} x^\tau + m_c^0$. Furthermore, if the series of mass inputs converges $\lim_{\tau \to \infty} \sum_{t=1}^{\tau} x^\tau = m_x^\infty$, then also the sum of cell states converges.*

*Proof.* Since $c_k^t \geq 0$, $x^t \geq 0$ and $m_h^t \geq 0$ for all $k$ and $t$,

$$|c_k^\tau| = c_k^\tau \leq \sum_{k=1}^{K} c_k^\tau = m_c^\tau \leq \sum_{t=1}^{\tau} x^\tau + m_c^0, \tag{19}$$

where we used Theorem 1. Convergence follows immediately through the *comparison test.* □

## D   ON RANDOM MARKOV MATRICES.

When initializing an MC-LSTM model, the entries of the redistribution matrix $\boldsymbol{R}$ of dimension $K \times K$ are created from non-negative and iid random variables $(s_{ij})_{1 \leq i,j \leq K}$ with finite means $m$ and variances $\sigma^2$ and bounded fourth moments. We collect them in a matrix $\boldsymbol{S}$. Next we assume that those entries get column-normalized to obtain the random Markov matrix $\boldsymbol{R}$.

**Properties of Markov matrices and random Markov matrices.**   Let $\lambda_1, \ldots, \lambda_K$ be the eigenvalues and $s_1, \ldots, s_K$ be the singular values of $\boldsymbol{R}$, ordered such that $|\lambda_1| \geq \ldots \geq |\lambda_K|$ and $s_1 \geq \ldots \geq s_k$. We then have the following properties for any Markov matrix (not necessarily random):

- $\lambda_1 = 1$.
- $\boldsymbol{1}^T \boldsymbol{R} = \boldsymbol{1}$.
- $s_1 = \|\boldsymbol{R}\|_2 \leq \sqrt{K}$.

Furthermore, for random Markov matrices, we have

- $\lim_{K \to \infty} s_1 = 1$ (Bordenave et al., 2012, Theorem 1.2)

For the reader's convenience we briefly discuss further selected interesting properties of random Markov matrices in the next paragraph, especially concerning the global behavior of their eigenvalues and singular values.

**Circular and Quartercircular law for random Markov matrices.**   In random matrix theory one major field of interest concerns the behavior of eigenvalues and singular values when $K \to \infty$. One would like to find out how the limiting distribution of the eigenvalues or singular values looks like. To discuss the most important results in this direction for large Markov matrices $\boldsymbol{R}$, let us introduce some notation.

- $\delta_a$ denotes the Dirac delta measure centered at $a$.
- By $\mu_{\boldsymbol{R}} = \frac{1}{K} \sum_{k=1}^{K} \delta_{\lambda_k}$ we denote the empirical spectral density of the eigenvalues of $\boldsymbol{R}$.
- Similarly we define the empirical spectral density of the singular values of $\boldsymbol{R}$ as: $\nu_{\boldsymbol{R}} = \frac{1}{K} \sum_{k=1}^{K} \delta_{s_k}$.
- $\mathcal{Q}_\sigma$ denotes the quartercircular distribution on the interval $[0, \sigma]$ and
- $\mathcal{U}_\sigma$ the uniform distribution on the disk $\{z \in \mathbb{C} : |z| \leq \sigma\}$.

Then we have as $K \to \infty$:

- **Quarter cirular law theorem:** (Bordenave et al., 2012, Theorem 1.1): $\nu_{\sqrt{K}\boldsymbol{R}} \to \mathcal{Q}_\sigma$ almost surely.
- **Cirular law theorem:** (Bordenave et al., 2012, Theorem 1.3): $\nu_{\sqrt{K}\boldsymbol{R}} \to \mathcal{U}_\sigma$ almost surely.

The convergence here is understood in the sense of weak convergence of probability measures with respect to bounded continuous functions. Note that those two famous theorems originally appeared for $\frac{1}{\sqrt{K}}\boldsymbol{S}$ instead of $\sqrt{K}\boldsymbol{R}$. Of course much more details on those results can be found in Bordenave et al. (2012).

**Gradient flow of MC-LSTM for random redistributions.** Here we provide a short note on the gradient dynamics of the cell state in a random MC-LSTM, hence, at initialization of the model. Specifically we want to provide some heuristics based on the arguments about the behavior of large stochastic matrices. Let us start by recalling the formula for $\boldsymbol{c}^t$:

$$\boldsymbol{c}^t = (\boldsymbol{1} - \boldsymbol{o}^t) \odot (\boldsymbol{R}^t \cdot \boldsymbol{c}^{t-1} + \boldsymbol{i}^t \cdot x^t). \tag{20}$$

Now we investigate the gradient of $\|\frac{\partial \boldsymbol{c}^t}{\partial \boldsymbol{c}^{t-1}}\|_2$ in the limit $K \to \infty$. We assume that for $K \to \infty$, $\boldsymbol{o}^t \approx \boldsymbol{0}$ and $\boldsymbol{i}^t \approx \boldsymbol{0}$ for all $t$. Thus we approximately have:

$$\|\frac{\partial \boldsymbol{c}^t}{\partial \boldsymbol{c}^{t-1}}\|_2 \approx \|\boldsymbol{R}^t\|_2. \tag{21}$$

$\boldsymbol{R}^t$ is a stochastic matrix, and $s_1 = \|\boldsymbol{R}^t\|_2$ is its largest singular value. Theorem 1.2 from Bordenave et al. (2012) ensures that $\|\boldsymbol{R}^t\|_2 = 1$ for $K \to \infty$ under reasonable moment assumptions on the distribution of the unnormalized entries (see above). Thus we are able to conclude $\|\frac{\partial \boldsymbol{c}^t}{\partial \boldsymbol{c}^{t-1}}\|_2 \approx 1$ for large $K$ and all $t$, which can prevent the gradients from exploding.

