# OpenReview forum: "MC-LSTM: Mass-conserving LSTM"
_ICLR.cc/2021/Conference — Reject_

### Official Review · AnonReviewer4 · 2020-10-14
**Nice idea, extensive experiments with clear references; having concerns about the architecture generalisation.**

**Rating:** 7
**Confidence:** 3

**Review:**

#### Summary

The authors proposed a new family of LSTMs (i.e. MC-LSTMS) which can be shown to have a mass conservation property for the LSTM memory cells. They have shown in various applications that these models lead to on par or better performance compared to state of art approaches. However, after examining the paper, I am not yet fully convinced that 1) a general unified MC-LSTMs can achieve good performance in all the mentioned cases 2) the conserved mass is interpretable (and corresponds to the problem’s invariant, conserved quantity). I have listed those concerns in the question sections as authors might have precise answers.

#### Pros

The authors in this paper have proposed a new family of LSTMs (i.e. MC-LSTMs) which can be shown to have a conservation property for the mass held by the LSTM memory cells. The authors have applied their models in three different prediction tasks involving quantity conservation, namingly summation(and subtraction), traffic prediction and rainfall runoff modelling. The proposed models (with some variants) achieve better or at least on par with current state of art models.

These different applications illustrate the usefulness of the models; the authors also reference appropriately to current state of art approaches, helping reviewers to well situate the paper’s contribution.

#### Questions:

Well I understand there are differences for each application (motivating to choose different neural architectures), it would be very interesting to know the performance of one quite general architecture (compared to say standard LSTM implemented in torch). In the paper, we have seen in the experiments:
- time - independent R^t used in arithmetics
- time - dependent R^t used in hydrology experiments
- Hypernetwork based R^t used in pendulum experiments
And also they vary on the auxiliary/mass inputs choices.
What would be the performance if the authors use a quite general architecture (for example the second setting) for all the experiments please?

Just for confirmation, in the experiments, the LSTMs involve forgetting gate right (contrary to what described in eq (1))

I don’t find explanations for the r value for the hydrology experiments. Meanwhile the r value for the traditional r value is different from what is chosen for the MC-LSTM. Why is this please?

#### Minor issues:

There seems some formatting issues at the end of “Mechanisms beyond storing are required for real-world applications” in Introduction.

In related work, Hamiltonian approaches (Greydanus et al. 2019) don’t seem to assume access knowledge to time derivatives w.r.t. Inputs. Rather, it favours the conservation parametrised by the Hamiltonian.

#### Minor suggestions:

In abstract, “expressed through continuity equations”, these aspects don’t seem to be addressed by MC-LSTMs, so I propose to not include this phrase in the abstract.

I suggest adding some citations In the introduction where the authors said “LSTM to excel at speech, text, and language tasks” http://nlpprogress.com/.

The authors mention in the abstract and introduction about the interpretability but only show the mass interpretation in the hydrology experiments. For which I would suggest the authors show it in at least one other experiments (e.g. traffic) to 1) be more convincing 2) highlight this property (which may reveal to be very beneficial for production systems)

I would be good to mention forgetting gate around equation (1) as those are used for later experiments in the paper.

#### Thank you for the authors to having answered my questions and having addressed all my comments. My main concern was about the generalisation of the proposed architecture and the results at the current revision are convincing to me. I believe this direction of the research together with the approach taken make a nice contribution to the conference. In consequence, I have raised my score from 5 to 7.

---

> ### Author Response · Authors · 2020-11-18
> **Response to reviewer 4**
>
> We thank the reviewer for this valuable and insightful feedback.
>
> Regarding Questions:
>  - (Redistribution matrix) The reviewer correctly pointed out the different versions of the R^T usage and provided an interesting proposal. We see MC-LSTM as a modular system and the empirical analysis also as a form of showcase, where the different implementations emphasize the flexibility of MC-LSTM. Thus, it is our opinion that the choice of R --- in size and behavior --- is also a form of inductive bias. We do agree with the reviewer that this was not clear in the original version. In the revised manuscript, we emphasize this point to make it  clearer to future readers. Regardless of that, we re-ran experiments of neural arithmetic problems, using the time-dependent formulation of R^t. We found that the results changed only slightly, which can be seen as an indicator that the architecture generalizes well.
>  - (Forget gate)  The reviewer is right. We used the original LSTM formulation by Hochreiter in Eq. (1), but for the experiments we used standard LSTMs that also involve a forget gate. In Eq. (1), we just want to explain our point of departure to motivate the MC-LSTM. We now state this clearer in the manuscript.
>  - (R value): We apologize for the unclarity of the statement with the r value. This “r” value is the Pearson correlation. There might also be some misunderstanding about the hydrology models for rainfall-runoff modeling and the models for snow-water equivalent. We reformulated the paragraph to improve clarity and avoid misunderstandings.
>
> Regarding Minor Issues:
>  - (formatting issues) We reformatted the document and hope that the formatting issue disappeared.
>  - (Hamiltonian approaches) Thanks, we reformulated this statement about Hamiltonian NNs in the related works section.
>
> Regarding Minor Suggestions:
>  - (continuity equations) We removed this statement in the revised manuscript.
>  - (additional NLP references) We added references as proposed.
>  - (Interpretability) We agree with this suggestion and added an analysis of trained MC-LSTM arithmetic tasks as well as for the pendulum (see also AnonReviewer1).
>  - (Forget gate) Good point, this should definitely be mentioned. We now state this in the experimental section.

---

### Official Review · AnonReviewer1 · 2020-10-25
**MC-LSTM is an interesting work.**

**Rating:** 6
**Confidence:** 3

**Review:**

<Summary>
1. Many real-world systems follow conservation laws.
2. This paper incorporates conservation laws into the RNN as an inductive bias.
3. This paper proposes MC-LSTM and shows that MC-LSTM follows conservation laws empirically and theoretically.
4. MC-LSTM utilizes a positive left-stochastic matrix to redistribute mass.
5. This paper validates the MC-LSTM on arithmetic tasks, traffic forecasting, pendulum, rainfall tasks.

<Strengths>
1. Incorporating mass conservation laws into the neural network is important.
2. This paper proposes MC-LSTM, a simple and effective mass-conserving model.
3. The motivation of the research and the proposed methods are straightforward.
4. Related works section provides a comparison between MC-LSTM and others such as the Markov chain, and it is also very interesting.

<Weaknesses>
1. This paper shows extensive experimental results, but there are less qualitative results.
2. This paper missing some important related works, such as a physics-guided recurrent neural network model (PGRNN)

<Questions and Additional Feedback>
1. Is there empirical (or theoretical) analysis for long-term gradient vanishing?
2. I wonder how MC-LSTM actually works. Is there qualitative analysis for a, i, o, and R?
3. What is the reason that Eq. (5) and Eq. (6) utilize L1 norm?
4. Does MC-LSTM can be extended to represent (2*mass) or (mass^2)-conserving properties?
5. MC-Transformer will be one of a good extension.

<Missing Reference>
1. Lagrangian Neural Networks
2. Physics-Guided Machine Learning for Scientific Discovery: An Application in Simulating Lake Temperature Profiles
3. Discovering physical concepts with neural networks

<Typos>
1. One of the greatest success stories of deep learning are => One of the greatest success stories of deep learning is
2. The former represent => The former represents

<After Rebuttal>
Thank you for your detailed response.
I will keep my positive score because my concerns are resolved partially.

---

> ### Author Response · Authors · 2020-11-18
> **Response to reviewer 1**
>
> We thank the reviewer for the positive comments and for the suggestions, which helped us to improve the manuscript. The main concerns of the reviewer are regarding the relative brevity of the qualitative analysis. We hope that the concerns are addressed in the revised manuscript. In the following, we provide answers to the individual concerns point-by-point:
>
> Regarding Weaknesses:
>  1. (Qualitative Results) We provide a qualitative analysis of the results for the arithmetic task and for the pendulum in the respective sections in the main paper and in the appendix.
>  2. (Related work) We added the missing references that the reviewer suggested.
>
> Regarding Questions & Additional Feedback
>  1. (Vanishing gradient) We thank the reviewer for this question. The gradient flow is largely governed by the redistribution matrix R. We would like to note that with a redistribution matrix close to identity, the gradient flow is similar to that of LSTM with a forget gate. For random redistribution matrices,  the circular law theorem for random Markov matrices can be used to analyze gradient flow. We now provide more details on this topic in the main manuscript (“Initialisation and gradient flow” in section 3) and in Appendix Section D.
>  2. (Qualitative analysis) This is a good suggestion. We conducted such an analysis for the arithmetic units and the pendulum that is now presented in the respective sections in the main paper and in the appendix.
>  3. (L1 norm): The L1 norm is not intrinsically necessary. The actual norm used should not have major influence on the results. We have two reasons for using the L1-norm here: First, we want to avoid saturation of the gates through large values of the cell states (which could also be achieved by other normalization methods). Second, L1 normalization supplies probability vectors that represent the current distribution of the mass across cell states which fits well with the stochastic matrices used for redistribution. In the revised manuscript, we extended the explanation for our choice in the corresponding section in the main paper.
>  4. (Extension) When the mass is directly supplied to MC-LSTM as mass input, quantities like mass^2 can not be represented in MC-LSTM. However, if mass inputs are fed into an MLP and then into MC-LSTM, other conserved quantities, such as 2\*mass or mass^2, can potentially be represented. Another option is to have an linear layer (or MLP) mapping the MCLSTM output to the output neuron, which could learn a scaling (such as 2\*mass). The fact that these strategies can work is shown in our experiments with neural arithmetics, where the output has to be scaled (recurrent arithmetic), and in MNIST-addition, where the quantity to be conserved has to be learned via CNN filters.
>  5. (MC-Transformer) We think this is an interesting idea. Generally, it does not seem to be a trivial extension because of the inherent mechanism of a transformer. However, we can point out that there is some relation to Transformer: each mass accumulator learns to attend to all other mass accumulators of the previous timestep via the redistribution matrix. Thus, a mass accumulator updates its own representation based on its attending to other mass accumulators, which is similar to how sequence tokens update their representations in Transformers and BERT.
>
> We also addressed the typos in the revised manuscript.

---

### Official Review · AnonReviewer2 · 2020-10-26
**Recommendation to Accept**

**Rating:** 7
**Confidence:** 3

**Review:**

The paper provides an interesting and novel LSTM structure named MC-LSTM, which extends the inductive bias of LSTM to deal with some real-world problems limited by conservation laws. The authors do some experiments related to traffic forecasting and hydrology to illustrate the effectiveness of MC-LSTM.  The new architecture is well-suited for predicting some physical systems, which is valuable.
##########################################################################

Reasons for score:

Overall, I vote for accepting.
I deem that the novel architecture based on LSTM that conserves quantites is useful and interesting.
My major concern is about some explanation about definitions and some additional ablation models (see cons below).
Hopefully the authors can address my concern in the rebuttal period.

##########################################################################Pros:

1. The paper takes one of the most important issue of some real-world systems:  conservation laws, which is important and
should be expressed through LSTM units.

2.  This paper provides comprehensive experiments, including both qualitative analysis and quantitative results, to show the effectiveness of the proposed LSTM.
The entire structure is organized well and the formulas are very detailed.

##########################################################################

Cons:

1. In Basic gating, the formula about input, why do you use softmax operator? Because in basic LSTM, there is sigmoid.
 It would be better to provide more details about it.

2. What are the advantages of MC-LSTM in terms of speed and resource consumption?
It would be more convincing if the authors can provide more cases in the rebuttal period.

##########################################################################

Questions during rebuttal period:

Please address and clarify the cons above

---

> ### Author Response · Authors · 2020-11-18
> **Response to Reviewer 2**
>
> We thank the reviewer for this critique and the statement about the fact that mass conservation is one of the most important issues for dealing with real-world systems. We agree with that. The two shortcomings the reviewer pointed out, were that the usage of the softmax is unclear and that the computation cost was not discussed in the original manuscript. As our individual answers below show, we took both points at heart and incorporated them in the updated version:
>
> Regarding Cons:
>  1. (Softmax) We thank the reviewer for pointing this out. Our aim was to give a basic formulation of MC-LSTM that provides the foundation for applying it to different tasks. However, it is possible to use other approaches that guarantee column-normalized matrices, e.g. by using sigmoids and normalization. For example, in the hydrology experiment we use normalized sigmoids as activation function in the input gate. We did not communicate this well enough. For the final version, we made sure that this flexibility is well-emphasized and better described in the manuscript.
>  2. (Computational Complexity) This is an interesting aspect that we now consider in more depth in the main paper. Indeed, the computational complexity of MC-LSTM is usually higher than for standard LSTMs. For a small number of mass inputs and time-independent MC-LSTM gates, the computational complexity is very similar. We now added a paragraph on “Computational complexity” in Section 3 of the main paper.

---

### Official Review · AnonReviewer3 · 2020-10-28
**The experimental evaluation shows that the proposed method is great at dealing with situations where conservation is required, which may be particularly important for real-world scenarios**

**Rating:** 7
**Confidence:** 3

**Review:**

In this paper the authors propose a novel architecture, called Mass-Conserving LSTM (MC-LSTM) based on LSTM. The authors base their work over the hypothesis that the real world is based over conservation laws related to mass, energy, etc. Thus, they propose that also the quantities involved in deep learning models should be conserved. To do so, they aim at exploiting the memory cells of the LSTM as mass accumulators and then force the conservation laws via the model equations. The authors finally show successfully the potential of this novel network into three experimental settings where several types of “conservation” are required (e.g. mass conservation, energy conservation, etc).

Pros:
+ they deal with a problem which can arise in non-laboratory scenarios in a novel way. Moreover, their results show that their method is great at dealing with situations where conservation is required, which may be particularly important for real-world scenarios
+ the paper explains in-depth all the decisions made, and it is well written. Moreover, I found really interesting how they deal with the related work and special cases. It shows a really in-depth understanding of up-to-date literature in the field.

Cons:
- (minor) the authors focus their experimental section to settings where mass (or energy, etc) conservation is required. It would be interesting to see how it performs also in settings where it is not required as well, thus showing whether this method also generalizes to different settings.

------

Authors' response addressed properly my request.

---

> ### Author Response · Authors · 2020-11-18
> **Response to Reviewer 3**
>
> We thank the reviewer for this positive assessment. We find it motivating that others also value solutions that focus on applicability of real-world applications.
>
> Regarding Cons (experimental settings):
>  - It is true that we focus on applications where conservation is part of the underlying process. We do not envision MC-LSTM as a general purpose network, which would, for example, replace LSTM in application domains such as next-word prediction, or similar. Rather we see it as a variation that is specifically tailored for applications where mass conservation (in our general sense) is helpful or necessary, which --- as the reviewer points out --- are many real-world systems.
> Although it is not impossible, it would also not be trivial to decide what the mass or auxiliary inputs in e.g. a language modeling task should be. Ultimately, we believe that some restrictions in terms of applicability will always apply to methods with greater inductive bias such as the MC-LSTM. However, within these confinements quite some flexibility is possible.
> We also want to point out that e.g. the hydrology example does not strictly require mass-conservation since the mass balance is not closed (i.e., only a part of the incoming water reaches the measurement station within the river, the remainder is “lost” due to evapotranspiration and percolation effects) and the incoming amount of water (i.e., the precipitation) is a noisy estimate. Other options for relaxing the setup exist too. For example,  one possibility would be to learn additional scaling factors for the inputs to account for systematic input-biases.

---

### Author Response · Authors · 2020-11-18
**Summary of changes to our manuscript**

We thank the reviewers for their positive assessment and insightful comments on our work. As the reviewers point out, we focus on scenarios where mass conservation applies, which directly links the research to many real-world scenarios. Indeed, we think that such scenarios are rather frequent. The reviewers rightly point out that this approach has innate advantages and disadvantages, which led us to the application domains that we present in the empirical section. We improved the manuscript with respect to clarity, and added qualitative analyses for the modelling tasks. Furthermore, we show that the more general time-dependent version of the redistribution matrix generalizes to situations in which time-independent dynamics are sufficient, and give detailed answers to the reviewer’s questions.

The main changes to the manuscript are:
 - We added an additional section to Appendix B that adds additional experiments with MC-LSTM in its most general formulation (i.e., with a time-dependent redistribution matrix) and compared against our previous results with the time-independent redistribution matrix for the arithmetic tasks.
 - We added qualitative analyses of MC-LSTM trained on different arithmetic tasks and the pendulum (see Appendix B.1.5 and Appendix B.3.1).
 - We added a paragraph on gradient flow in Section 3 of the main paper and section D in the appendix.
 - We added a paragraph on the computational complexity of MC-LSTM in the Section 3 of the main paper.
 - We added the references pointed out by the reviewers.
 - We reformulated various sentences/sections according to the suggestions of the reviewers to clarify points that were previously unclear.

---

### Decision · Program_Chairs · 2021-01-07
**Final Decision**

**Decision:**

Reject

**Comment:**

The paper proposes a variant of recurrent neural networks based on Long Short-Term Memory. Unlike the standard LSTM, the proposed mass-conserving LSTM subtracts the output hidden state of the LSTM from its current cell state, thus preserving the "mass" stored in the cell states at each step. A left-stochastic recurrent weight matrix is also used to conserve the "mass" across the time steps. Empirical experiments demonstrated the effectiveness of the proposed MC-LSTM on a range of datasets such as addition & arithmetic tasks, traffic forecast, and rainfall modeling models.

Several issues were clarified during the rebuttal period in a way that satisfied the reviewers. However, some concerns still remain unanswered:

1) This is an empirical paper that proposes a modified LSTM that brings forward a few different ideas: L1 norm, stochastic transition matrices, and subtracting the output hidden states. An ablation study is a MUST in such an applied work. It has been pointed out by other reviewers that there are many prior references on LSTMs variants. It would greatly strengthen the paper by considering more diverse baselines. There is no experiment nor discussion on how much each modification helps wrt the final accuracy. Thus it remains unclear how the results can generalize to other problems.

2) Although the results seem convincing across various datasets that mass conservation seems to help, the datasets are non-standard benchmarks in the machine learning conferences thus there is a lack of competitive prior baselines. As the proposed LSTM has a different number of parameters compared to the standard LSTM, is it fair to compare the different architectures under the same number of neurons? What happens if we compare the architectures with the same number of parameters? And how well does the model scale as we vary the hidden size? It would be helpful to keep the contributions into perspective by using standard RNN benchmark datasets such as Penn TreeBank or Wiki-8.

Overall, the basic idea seems interesting, but the lack of ablation studies significantly hurt the contribution and the positioning of the paper. Given the current submission, the paper needs further development, and non-trivial modifications, to be broadly appreciated by the machine learning community.

---

> ### Comment · ~Günter_Klambauer1 · 2021-02-04
> **Response to AC decision**
>
> We accept and at the same time regret the area chair’s decision.
>
> Ad 1.) We completely agree that ablation studies are a MUST to investigate the influence of architectural choices. We had already investigated this in the appendix sections and during method development and we will add a larger ablation study in the main paper. On the other hand, the architectural choices were based on a single idea of introducing mass conservation. This idea directly implied the design of the input gate, redistribution matrix and output gate. We will elaborate on how these architectural components influence the final performance of MC-LSTM.
>
> Ad 2.) We completely agree that prior competitive baselines are necessary for such works. We therefore benchmarked MC-LSTM against several baselines and recently published methods, for example different LSTM variants, neural arithmetic units, and neural arithmetic logical units in the area of neural arithmetics. We extend our comparative effort by adding physics-inspired architectures, such as Hamiltonian neural networks. Since in standard benchmarks for language models there is no quantity to conserve, we used benchmarking frameworks of areas in which a physical quantity has to be conserved. We evaluated MC-LSTM on the benchmarking setting in the area of neural arithmetics (Madsen, 2020), in physics on the damped pendulum modeling task by Iten et al. (2020), and in environmental modeling on flood forecasting (Kratzert et al., 2019). We emphasize that MC-LSTM is currently the best neural arithmetic unit and the best model for predicting peak flood events, which can strongly impact environmental modeling.